# Machine learning and deep learning-based approach to categorize Bengali comments on social networks using fused dataset

Khandaker Mohammad Mohi Uddin[1]*, Hasibul Hamim[2]*, Mst. Nishat Tasnim Mim[2]*, Arnisha Akhter[3], Md Ashraf Uddin[4]

1 Department of Computer Science and Engineering, Southeast University, Dhaka, Bangladesh,
2 Department of Computer Science and Engineering, Dhaka International University, Dhaka, Bangladesh,
3 Department of Computer Science and Engineering, Jagannath University, Dhaka, Bangladesh, 4 School of Information and Technology, Deakin University, Geelong Warun Ponds, Australia

* jilanicsejnu@gmail.com (KMMU); hasibulhamim2020@gmail.com (HH); mimkst2000@gmail.com (MNTM)

**Data Availability Statement:** The investigation has been reinforced through data previously released, and these are accessible online. https://www.kaggle.com/datasets/cypher1337/dataset-for-

## Abstract

Through the advancement of the contemporary web and the rapid adoption of social media platforms such as YouTube, Twitter, and Facebook, for example, life has become much easier when dealing with certain highly personal problems. The far-reaching consequences of online harassment require immediate preventative steps to safeguard psychological well-ness and scholarly achievement via detection at an earlier stage. This piece of writing aims to eliminate online harassment and create a criticism-free online environment. In the paper, we have used a variety of attributes to evaluate a large number of Bengali comments. We communicate cleansed data utilizing machine learning (ML) methods and natural language processing techniques, which must be followed using term frequency and reverse document frequency (TF-IDF) with a count vectorizer. In addition, we used tokenization with padding to feed our deep learning (DL) models. Using mathematical visualization and natural language processing, online bullying could be detected quickly. Multi-layer Perceptron (MLP), K-Nearest Neighbors (K-NN), Extreme Gradient Boosting (XGBoost), Adaptive Boosting Classifier (AdaBoost), Logistic Regression Classifier (LR), Random Forest Classifier (RF), Bagging Classifier, Stochastic Gradient Descent (SGD), Voting Classifier, and Stacking are employed in the research we conducted. We expanded our investigation to include different DL frameworks. Deep Neural Networks (DNN), Convolutional Neural Networks (CNN), Convolutional-Long Short-Term Memory (C-LSTM), and Bidirectional Long Short-Term Memory (BiLSTM) are all implemented. A large amount of data is required to precisely recognize harassing behavior. To rapidly recognize internet harassment written material, we combined two sets of data, producing 94,000 Bengali comments from different points of view. After understanding the ML and DL models, we can see that a hybrid model (MLP+SGD+LR) performed more effectively when compared to other models, its evaluation accuracy is 99.34%, precision is 99.34%, recall rate is 99.33%, and F1 score is 99.34% on multi-label class. For the binary classification model, we got 99.41% of accuracy.

cyberbully-detection-bengali-comments. {Accessed-Date: 4:23 PM Thursday, November 17, 2023 (GMT+6) Dhaka} https://www.kaggle.com/datasets/smnuruzzaman/facebook-sentiment-analysis-bangla-language. {Accessed-Date: 1:09 AM Thursday, November 20, 2023 (GMT+6) Dhaka} The following GitHub repository link contains the combined dataset file and the code for the models: https://github.com/hasibulhamim2020/PONE-D-24-03168__Machine-Learning-and-Deep-Learning-Based-Hybrid-Strategy.

**Funding:** The author(s) received no specific funding for this work.

**Competing interests:** The authors have declared that no competing interests exist.

**Abbreviations:** **AdaBoost**, Adaptive Boosting; **ANN**, Artificial Neural Network; **BERT**, Bidirectional Encoder Representations from Transformers; **BiGRU**, Bidirectional Gated Recurrent Unit; **Bi-LSTM**, Bidirectional Long Short-Term Memory; **CLSTM**, Convolutional Long Short-Term Memory; **CNN**, Convolutional Neural Network; **DNN**, Deep Neural Network; **DT**, Decision Tree; **ET**, Extra Tree; **IHT**, Instance Hardness Threshold; **KNN**, K-Nearest Neighbor; **LightGBM**, Light Gradient-Boosting Machine; **LR**, Logistic Regression; **MLP**, Multi-Layer Perceptron; **MNB**, Multinomial Naive Bayes; **NLP**, Natural Language Processing; **RF**, Random Forest; **RNN**, Recurrent Neural Network; **SA**, Sentiment Analysis; **SGD**, Stochastic Gradient Descent; **SN**, Social Networks; **SVM**, Support Vector Machine; **TF-IDF**, Term Frequency-Inverse Document Frequency; **XGBoost**, Extreme Gradient Boosting.

## Introduction

Social media platforms are vibrant platforms for communicating potential, inspiring ideas, stories, and important information. Due to the development of high-speed internet and communications technologies, a large number of people from all walks of life have joined social networks (SN) and share their opinions about a wide range of topics [1, 2]. Nowadays, a significant issue on social media is abuse or threats, which is known as cyberbullying. Social media platforms act as a robust medium for conversation [3]. It also helps in providing various facilities or encouraging ideas. As of July 2023, there will be 4.88 billion engaged social networking user identities worldwide, representing 60.6% of the world's population. There are 53 million Bangladeshi users of Instagram, Messenger, and Facebook that are part of Meta's global advertising audience. At the beginning of 2023, the percentage of the Bangladeshi population using the internet was 38.9 percent, with 66.94 million users. As of January 2023, 26.0 percent of Bangladesh's population, or 44.70 million people, were active on social media. As seen in "Fig 1", the total number of individuals who use (in millions) the social networking sites that are available in Bangladesh, Facebook (43.25), YouTube (34.4), Instagram (4.45), Messenger (20.35), LinkedIn (5.9), and Twitter (1.05) [4]. Because Bengali is the native tongue of Bangladesh, it is reasonable to predict that a sizable proportion of these people will use the Bengali language on social networking platforms. However, it is stated that there are about 228 million native speakers of Bengali globally, most of the Indian states associated with Tripura, Assam, and West Bengal, but also in significant numbers in the UK, the USA, and the Arabian Peninsula, with 160 million of them being Bangladeshis [5].

Businesses, governmental organizations, and event planners can fully comprehend people's sentiments and perceptions by analyzing the data obtained via SN. However, because of the exponential rise in SN users, data is abundant with a vast quantity of remarks and posts, making it difficult for people to precisely extract pertinent data from the texts [6]. To address human limitations, hidden and in-sight information must be automatically removed from online-generated text. As social media platforms keep growing, the frequency of cyberbullying is increasing [7]. People are regularly harassed by strangers and unauthorized users on social media platforms [8]. Nevertheless, the nation has seen an enormous rise in internet access in the past few years, coupled with a growing proclivity to use online platforms, such as social networking sites, for educational activities. As a result of this, the number of victims of cyberbullying has risen, leading to mental health problems between learners. According to recent research, 73.71% of cyberbullying victims are women [9]. According to a preferred "Bangladesh Cybercrime Trend 2023" published in 2022 through the Internet Crime Awareness Foundation (CAF), 52.21% of the total reported via the internet actions were related to cyberbullying and brutal social networking articles, in undergraduates making up nearly all of these those affected [10]. Cyberbullying harms individuals' mental health, resulting in higher instances of anxiety and sadness. Cyberbullying and cyberstalking affect people psychologically. Abusers use social media platforms' anonymity to their advantage, enabling their vicious behavior to go unpunished. Additionally, as harassment increases in frequency over time, things get worse [11].

For example, identifying abusers and making them responsible is more difficult than with the classic bully. Perpetrators employ phony IDs, labels, and locations to harass people using technological devices and services (social websites, phones, electronic mail, and more), and they may also utilize encrypted networks to conceal the truth about themselves and their location from others. Furthermore, because cybercrime happens via online tools and technology, it can reach vast audiences quickly, making it more harmful than face-to-face incidents. Additionally, bullying is a never-ending form of humiliation that leaves victims feeling helpless.

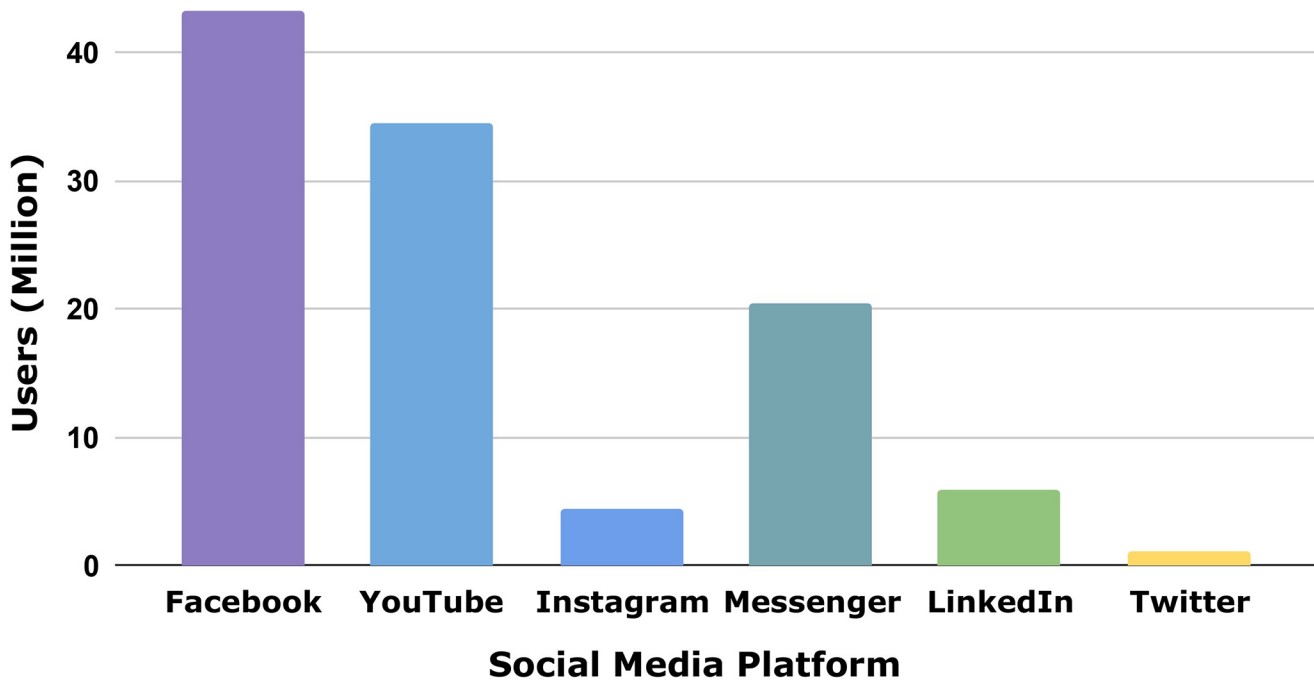

**Fig 1. Social media user distribution in Bangladesh (2023).**

That is why we require an automated solution to assist us in identifying and preventing the majority of bullying.

Plenty of investigations were conducted to recognize insulting posts and comments in both English and various languages [12, 13]. Additionally, there are only a few investigations that concentrate on Bengali language harassing identification [5, 11]. As a result, this research provides a possibility to contribute towards the identification of Bengali internet harassment and protect users from this kind of harassment. In Bengali, there are numerous methods for identifying cyberbullying. Their performance assessments for detecting bullying are insufficient for shielding people via Bengali cyberbullying. As a result, there is currently a pressing desire for a stronger approach geared toward detecting a variety of violent Bengali written material. Techniques that utilize ML, as well as DL, may prove highly successful in recognizing and eliminating violent Bengali material via social networking sites. To identify cyberbullying, it is essential to investigate and develop more complex, quick, and flexible detection techniques. By gathering the cessation of current methods that we operate regarding innovative approaches that can effectively impact the ever-changing step of cyberbullying.

Multiple components are created in our paper. We attempt to give the highlights of the associated study part in the field of study for identifying cyberbullying and the group of offensive text in the very first section. The composition of both datasets is described in the methodology section. We have also cleaned data based on a variety of parameters, including eliminating special characters, multiple spaces, punctuation marks, non-Bengali characters, and numbers, among others. Following data preprocessing, we extract features for ML classifiers employing TF-IDF transformation with Count-Vectorizer. On the other hand, tokenization with padding is used to feed our DL models. We establish a vocabulary with a capacity of over 20,000 ML feature extraction, which helps us with our large combined dataset. We define thoroughly several algorithms for ML and DL and implement them all via a comparable

arithmetic equation. To prepare for cyberbullying, we intend to talk about the initial stages, classifier selection, and setup, and the final implementation of each strategy. The findings of all approaches will be addressed and contrasted to identify the problems and constraints that were discovered. The conclusion of our paper includes a summary and evaluation of what our research means in this field as well as for the larger community in general. The main outcomes of the present study paper are as follows:

- A hybrid machine-learning method for detecting online harassment.

- Cleansed data according to several criteria, such as removing punctuation, numerous spaces, particular characters, non-Bengali text, and numbers.

- For the training and testing of the framework, two datasets are combined.

- Count-Vectorizer, along with TF-IDF for ML and tokenization with padding for DL models are used to extract the feature.

- For efficient undersampling, Iterative Hard Thresholding (IHT) was employed.

- K-Fold cross-validation was used to evaluate the model robustly.

- Tested for feature significance and model efficacy using ANOVA and Chi$^2$ tests.

- Examined the performance metrics between the ML, DL, and hybrid models to identify the most successful model.

- Created a web application that allows users to interact with the most optimal model.

- Suggested using a mixed machine-learning approach to identify online harassment.

## Related work

Multiple research projects were carried out in the area of identifying cyberbullying, alongside a particular focus on wellness for the Bengali language. The results of these studies show that there is a need for efficient ways to combat cyberbullying on social media platforms, which serves as a compelling argument for the current research. The relevant works on multi-class Sentiment Analysis (SA) in Bengali and English are covered in this section. We concentrated on important features of SA papers, including the quantity of datasets, number of classes, employed techniques, and outcomes. Different linguistic perspectives have emerged in response to multiclass SA in recent years.

With the introduction of a CNN and LSTM-based classifier that achieved 85.8% accuracy as well as 86% F1 scores upon a set of 42,036 comments on Facebook, Haque et al. [5] addressed the difficulties associated with multi-class analysis of sentiment in Bengali social media comments. By demonstrating its effectiveness in real-world sentiment detection through a web application integration, the suggested model outperforms baseline techniques.

However, utilizing an encoder-decoder-based LSTM network, Das et al. [11] proposed a different approach for identifying instances of hateful language in Bengali. To address the various classes hateful language issue, they employed TF-IDF, word2Vec, and a 1D CNN model in a network using LSTM. An encoder-decoder machine learning model a well-liked NLP tool was presented in this paper as a way to categorize feedback submitted by users in Bengali on profiles on Facebook.

Eshan et al. [12] also examined a variety of machine learning algorithms, including Random Forest, multinomial Naïve Bayes, Support Vector Machine with linear, Radial Basis Function

(RBF), Polynomial, and Sigmoid kernels. They also contrasted these algorithms with characteristics utilizing single-word, bigram, and trigrams, which are used by Count Vectorizer and Tfidf Vectorizer, but found that the SVM Linear kernel produced the most effective outcomes.

A Gated Recurrent Unit (GRU) model was developed by Ishmam et al. [13] to classify feedback from customers on social media sites. 5126 pieces of Bengali feedback were gathered for the purpose of research, and they were divided into five categories: political discourse, spiritual remarks, encouragement, insults, and discrimination towards race and religion. The GRU structure recognized hateful speech with an accuracy of 70.10%.

A binary and multilabel classifier algorithm was presented by Ahmed et al. [14] to recognize abuse statements on Facebook pages. 44,001 user reviews from well-known public Facebook pages were examined for this study and were divided into classes such as non-bully, sexual, threat, troll, and religious. This NN + Ensemble method produced a multilabel classification accuracy of 85% and a binary classification accuracy of 87.91%.

A model for identifying cyberbullying in texts written in Bangla and Romanized was developed by Ahmed et al. [15] using ML and DL techniques. Three social media datasets were produced by their research: one of them for Bangla, another for Romanized Bangla, and one combined dataset. In the combined dataset, the ML algorithm Multinomial Naive Bayes (MNB) achieved an accuracy rate of 80%.

A Bengali-language method for identifying cyberbullying on social media was proposed by Emon et al. [16]. They used 44,001 Bengali comments from Facebook to test different transformer models, such as Bengali BERT, Bengali DistilBERT, and XLM-RoBERTa. Out of all the models, the XLM-RoBERTa model had the highest accuracy rate (85%) and F1 score (86%).

A technique for using ML algorithms to recognize abusive language in Bangla was proposed by Mahmud et al. [17]. Using logistic regression (LR) and annotated translated Bengali corpora, they were able to identify bullying in Bengali with a 97% accuracy rate.

63,000 Bengali Facebook comments from various celebrity pages were compiled by Khan et al. [18] in order to group fans' sentiments toward the celebrity into five categories: happy, excited, upset, shocked, and content. The feature extractor they used to train SVM, NB, RF, KNN, and NN was TF-IDF. They used the SVM classifier to predict a person's attitude toward a celebrity with a 62% accuracy rate. Even though they employed a sizable dataset for their investigation, the dataset's class imbalance issue resulted in a low accuracy score.

The HS-BAN slanderous speech database in Bengali, which has more than 50,000 categorized statements, was made available by Romim et al. [19]. They investigated language features along with artificial neural network-based methods to develop a common detection of hateful speech systems for Bengali. These comparisons demonstrated that sentences incorporating algorithms developed on unofficial papers performed better than those developed on official texts, leading to the Bi-LSTM models outperforming Fast Text casual word implementation with an F1 score of 86.78%.

Utilizing machine learning techniques like NB, J48, SVM, as well as KNN, Akhter et al. [20] completed an analogous binary categorization assignment to identify Bengali cyberbullying comments. A collection of 2400 Bengali comments classified as either attacked or not was used in their tests. They used the TF-IDF as an extractor of features to teach the SVM classification algorithm, and they were able to achieve 97% accuracy. Nevertheless, multi-class SA was not present in the experiment because it tended toward categorization into binary categories.

In order to detect cyberbullying in Bengali on social media, Akhter et al. [21] developed a strong hybrid machine-learning model that achieved high accuracy rates of 98.57% and 98.82% in binary as well as multilabel identification. Effective text preparation, feature extraction with TFIDF, and dataset normalization with the instance hardness threshold are all part of their methodology.

**Table 1. Related publications along with boundaries.**

| References | Models | Dataset Size | Accuracy | Limitations |
|---|---|---|---|---|
| **Haque et al. 2023 [5]** | SVC, SGD, RF, LR, MNB, DT, CLSTM, BiGRU, BiLSTM, and LSTM | 42,036 | CLSTM: 85.80% | Detection Accuracy is lower. |
| **Das et al. [11]** | RNN, attention mechanism, LSTM, GRU, CNN | 7,425 | CNN: 77% | Detection Accuracy is lower and low dataset coverage. |
| **Eshan et al. [12]** | RF, MNB, SVM | 2500 | SVM: 75% | Limited accuracy and dataset coverage. |
| **Ishmam et al. [13]** | GRU, Adaboost, RF, NV, SVC | 5,126 | GRU: 70.10% | Limited accuracy and dataset coverage. |
| **Ahmed et al. 2021 [14]** | Random Forest, SVM, KNN, Naïve Bayes, Hybrid neural network (CNN-SVM) | 44,001 | CNN-SVM: 87.91% SVM: 85% | Limited to specific categories and dataset with limited accuracy. |
| **Ahmed et al. 2021 [15]** | MNB, SVM, LR, XGBoost, CNN, LSTM, BLSTM, GRU | 12000 | MNB: 80% | Limited accuracy and dataset coverage. |
| **Emon et al. 2019 [16]** | LinearSVC, LR, MNB, RF, ANN, RNN, LSTM | 4700 | RNN: 82.20% | The dataset used for the study was relatively small. Detection Accuracy is also lower. |
| **Mahmud et al. 2023 [17]** | LR, MB, DT, RF, SVM, AdaBoost, GB, SGD, ET, KNN, MLP | 3000 | LR: 97% | The dataset used for the study was relatively small. Limited preprocessing approach used. |
| **M. Khan et al. 2021 [18]** | SVM, NB, RF, KNN, NN | 63,000 | SVM: 62% | Detection Accuracy is also lower. Only 2% of data labeled for the "Religious" class |
| **Romim et al. 2021 [19]** | Bi-LSTM | 50,314 | Bi-LSTM: 86.78% F1-score | Limited to hate speech detection and Bengali language. Their study did not discuss the potential impact of dataset. |
| **Akhter et al. [20]** | NB, J48, SVM, KNN | 2,400 | KNN: 97.73% | Limited accuracy and dataset coverage. |
| **Akhter et al. 2023 [21]** | DT, RF, LR, and MLP | 44,001 | MLP: 98.82% | No DL models are used. Models' execution time is high. |

The previous research on discrimination recognition in multiple languages, including Bengali, abusive content identification, and cyberbullying detection is summarized here. These studies have provided insightful information. The research findings still have some shortcomings. A few demonstrated lower accuracy rates, underscoring the need for development, while others limited generalizability by concentrating on particular languages and text types. It was common to rely on certain ML algorithms and approaches, which necessitated investigating a larger variety of strategies. Inadequate rationale for classification decisions and a restricted number of dataset categories were also noted. Our research intends to create a strong hybrid machine-learning approach that covers harmful content categories, and Bengali languages to close these gaps. We will investigate different ML algorithms and DL approaches, offer thorough explanations for categorizations, and further the development of techniques for detecting and preventing cyberbullying. "Table 1" depicts prior research with boundaries. To overcome any required boundaries, we employed 8 various ML algorithms with 4 distinct DL models, as well as some ensemble approaches employing voting classifiers along with stacking to determine the most effective result from the best model. Our method processes huge quantities of data, assisting in the detection of bullying. It additionally includes five classes, each with distinctive positions, respectively.

## Materials and methods

The following section outlines our suggested approach in addition to the numerous ML and DL algorithms implemented in the structure. To begin, we will describe the way the concept operates. After that, the ML and DL algorithms are given a brief overview.

"Fig 2" depicts our proposed model's workflow. The proposed model is divided into eight major sections, the first of which is the collection of Bengali bullying text data. We began by

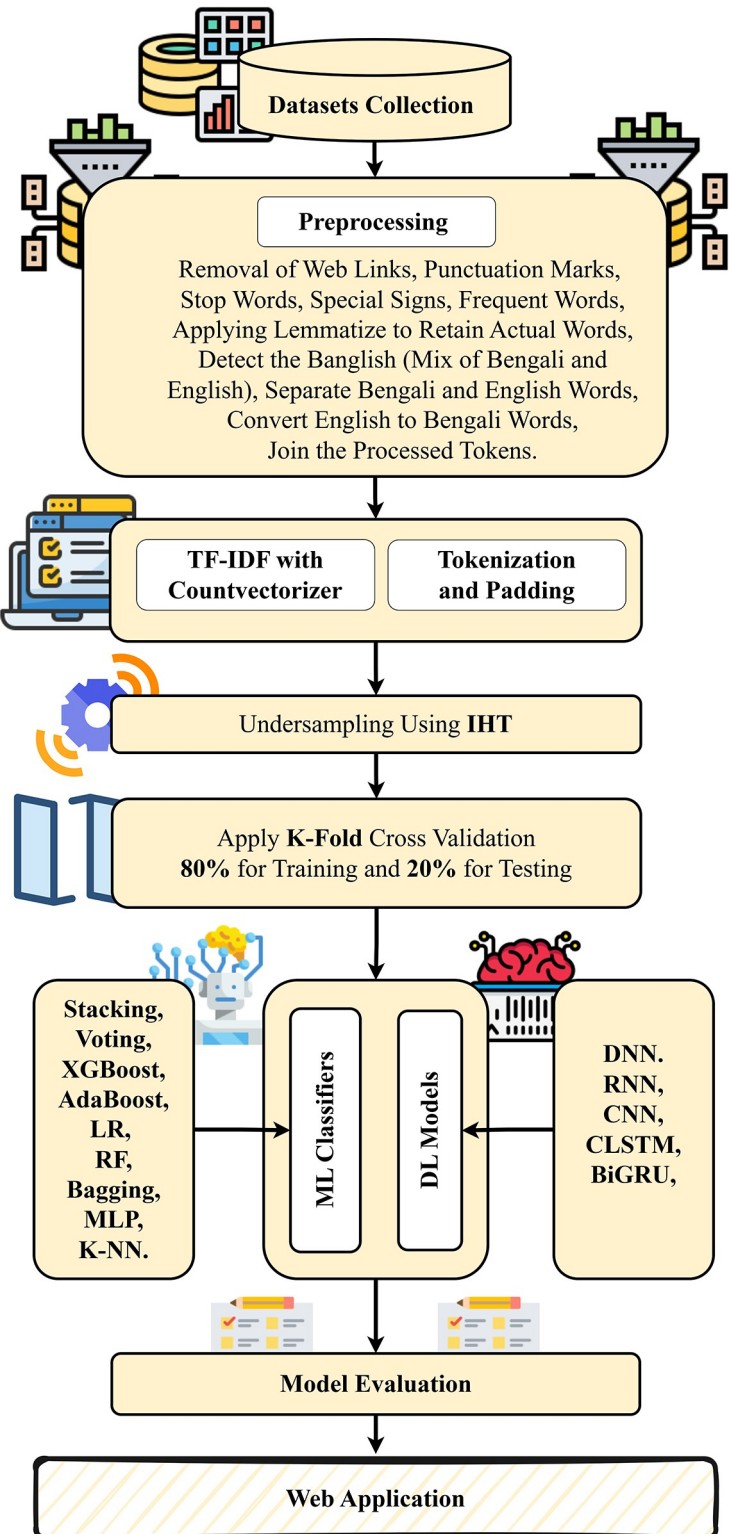

**Fig 2. Workflow of our proposed model.**

gathering information using two openly accessible databases of Bengali social media comments. Then we combined them into a single dataset that was separated into five distinct groups: trolls, religious, sexual, threats, and not bullying. The text was then preprocessed to remove websites, punctuation, digits, emojis, special signs, and Bengali stop words in order to prepare the data. To transform text data into numerical information, we use a count vectorizer with a TF-IDF transformer to build our ML classifiers and tokenization with padding to feed our DL models. We use IHT to resample the dataset for potential class imbalances during training and testing. To split the data, we used k-fold cross-validation and ML classifiers such as Staking, Voting, SGD, LR, RF, MLP, Bagging, XGBoost, Adaboost, and K-NN. For comparative analysis, DL models such as BiGRU, CNN, CLSTM, DNN, and RNN were also used. And assessed performance using traditional evaluation metrics such as accuracy, precision, recall, and F1 score. Finally, we developed a web application based on the best-performing models.

## Dataset collection

The lack of thorough assessment of publicly available databases in Bengali has become a significant disadvantage of the various classes of the Bengali sentiment analysis approach. The majority of present investigations depend on privately collected datasets, often focusing on diverse difficulties as well as including just a tiny Bengali writing the database. Initially, we gathered two sets of data from Kaggle [22, 23] over a multi-class cyberbullying analysis. We require additional understanding to get around previous research constraints, that are going to be gathered via the analysis of enormous quantities of knowledge. This is the reason in our line of work, that we integrate the data sets. We mixed them into a single MS Excel file featuring approximately 94,000 specimens, every single one labeled as Not Bully, Troll, Sexual, Religious, as well as Threat. After that, our team split the data into 20% for testing purposes and 80% for learning.

"Table 2" presents here an in-depth analysis of content categorization across various categories, with designations for each category like "Not Bully," "Troll," "Sexual," "Religious," as well as "Threat." In terms of total comments, the "Not Bully" class has the most with 33,579, after "Troll" with 23,193, "Sexual" with 18,026, "Religious" alongside 15,424, and lastly "Threat" with 3,778. When it comes to total words, the "Not Bully" category once again takes the lead with 390,688, next to "Troll" with 317,191, "Sexual" with 345,339, "Religious" with 392,196, and "Threat" with 60,269. "Fig 3" represents the percentage of comments for each class. This knowledge is essential for comprehending the makeup and features of various content classifications, laying the groundwork for additional research and information, specifically in the realms of natural language processing (NLP) as well as compromise. "Fig 4" shows the word cloud of our combined dataset.

The labels in the dataset are described below:

- Not Bully: indicates a group of comments which are not related to bullying.

- Troll: This class represents comments that have been identified as troll-related.

- Sexual: This class represents comments that have been recognized to contain sexual content.

**Table 2. Numeric features of the dataset.**

| Class Names | Not Bully | Troll | Sexual | Religious | Threat |
|---|---|---|---|---|---|
| Total Comments | 33,579 | 23,193 | 18,026 | 15,424 | 3,778 |
| Total Words | 3,90,688 | 3,17,191 | 3,45,339 | 3,92,196 | 60,269 |
| Unique Words | 37,667 | 33,533 | 33,820 | 32,257 | 9,047 |

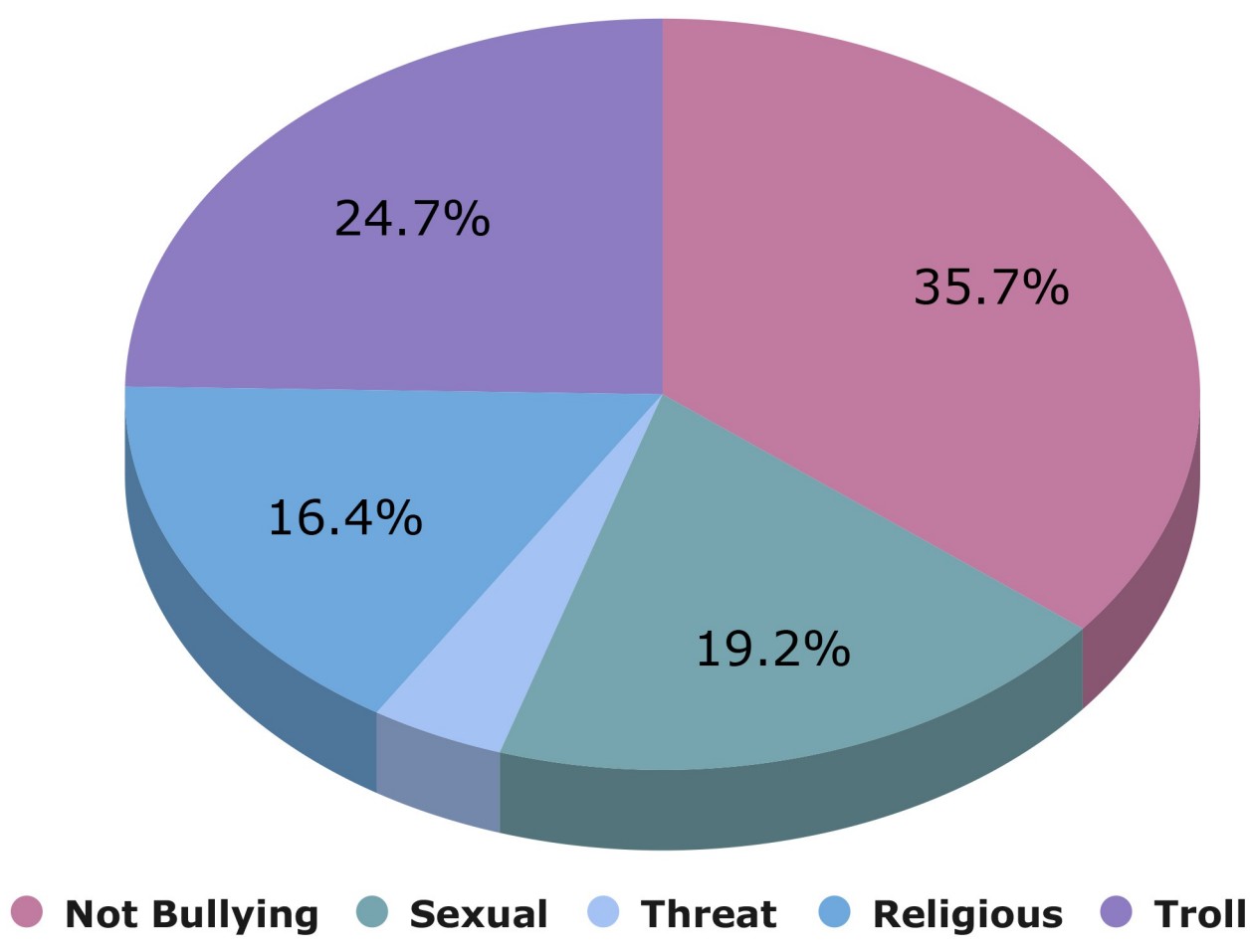

**Fig 3. Percentage of comments for each class.**

- Religious: This class represents comments that have been discovered to contain religious content.

- Threat: This class represents comments that have been recognized as containing threats.

### Data preprocessing

The preprocessing of data has become the procedure of preparing unprocessed data and over-fitting a model. That constitutes one of the absolute most essential processes within the development of a framework. We manage empty strings substituting via technique to be it is categorized characteristics, every web link, commas, as well as unique signs within the content preliminary analysis. The content is subsequently converted to tokens, and every Bengali stop word and common word is removed. Following that, we used lemmatization to keep the text's actual phrases. In addition, we manage Banglish sentences, which include a combination regarding Bengali as well as English sentences, by first spotting the Bengali language, then splitting the Bengali as well as the English language, transforming the English into Bengali words, and at last collaborating with the resultant tokens of value. This cleansing helps to reduce input depth, preserve important details, and prepare knowledge for building models

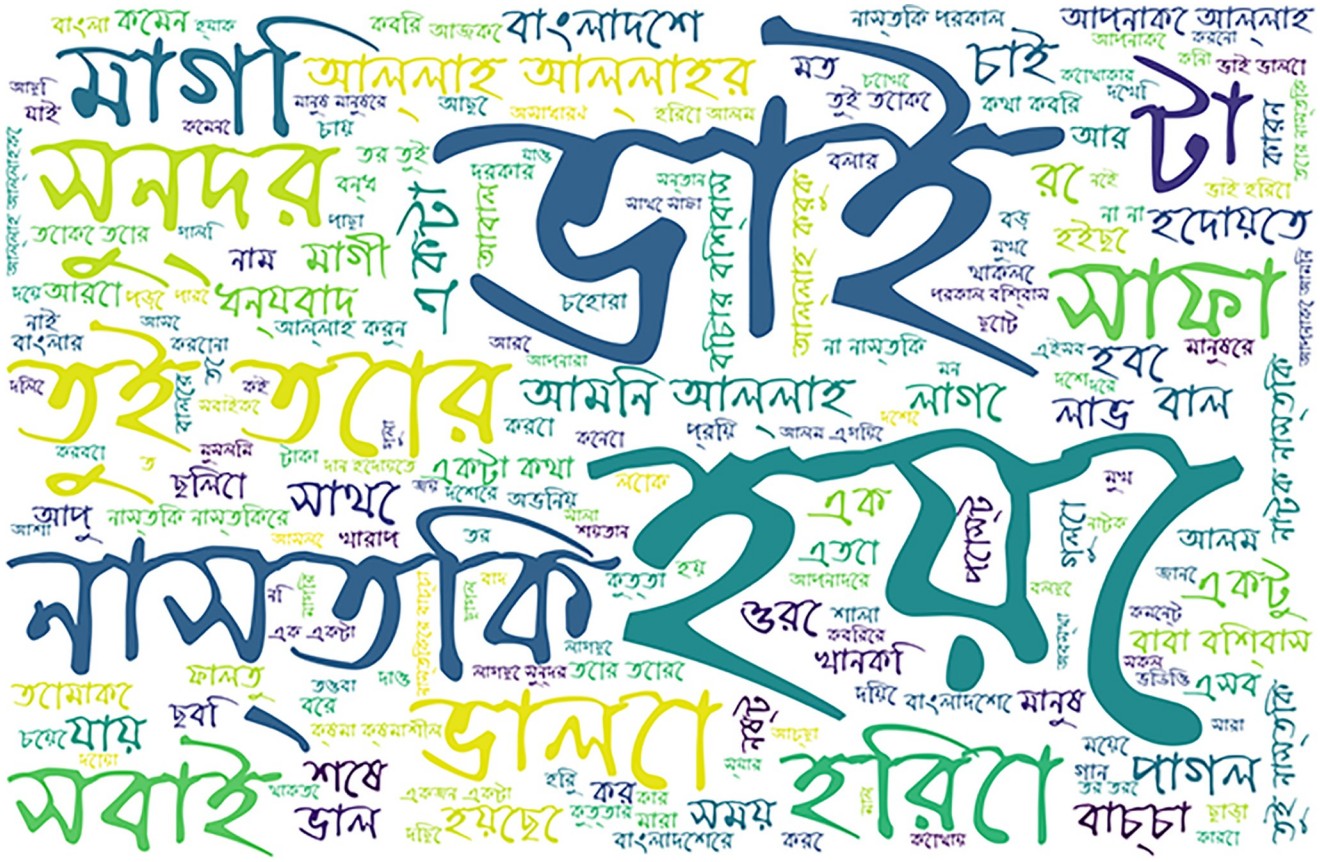

**Fig 4. Word cloud of dataset.**

with little difficulty [24]. "Fig 5" shows the density of comments length before and after pre-processing the dataset.

From this plot, we discover that before preprocessing the dataset, there were 210 words present in a sentence, most of the sentences contained 0 to 30 words and approximately 10 words contained the most frequently occurring sentences. On the other hand, after completing our desired preprocessing approach, we utilized the fact that the word length for sentences decreased to 150. Here also, we see that approximately 8 words are most frequent for the sentences, and at this stage, density is increased compared to before preprocessing. "Fig 6" shows each column contains the original comments before preprocessing and processed comments after preprocessing, along with the processed comments translated form of the Bangla dataset.

### Feature extractions techniques

The extraction of features is a dimensionality-reducing technique employed in ML that visualizes higher-dimensional participation through a variety of low-dimensional sets of attributes. The effectiveness of the ML model may be substantially enhanced, and the amount of computing work may decrease once pertinent characteristics are taken out [25, 26]. In the identification of text, every ML and DL classifier treats writing words as a number instead of natural language content. Before employing any algorithm, natural language terms must be transformed to vector style, enabling text sentences to be symbolized using vector representations.

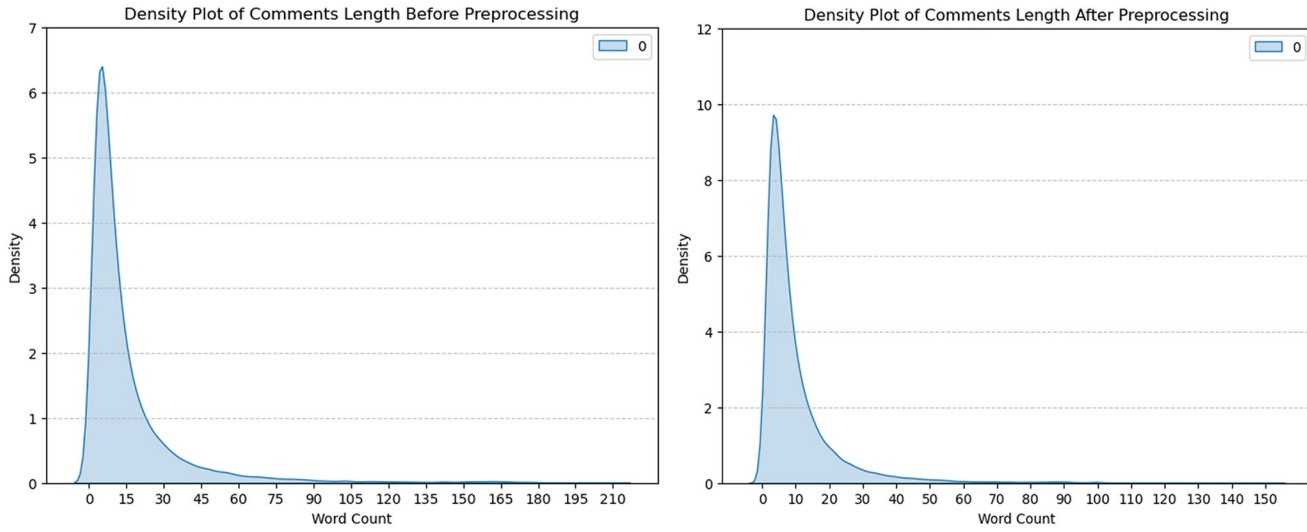

**Fig 5. Density plot of comments length before and after preprocessing the dataset.**

TF-IDF, CV, and tokenization methods are able to be employed to transform every single one of the vocabularies into a vector of features. The aforementioned methods may record the randomized visualization of sentences in an already established vector domain, just like vectors with real values [27]. The dispersed vector format for words implies vectors can be effectively represented within a low-dimensional space for features, which proved beneficial in prior multi-class categorization investigations. In the present research, methods for feature extraction were employed in two steps: ML classifiers were trained using TF-IDF with CV, and DL models were trained using tokenization and padding.

**Feature extractions for machine learning classifiers.** We combine the technique Count Vectorizer (CV) with TF-IDF [28] to create a powerful text data processing algorithm. CV tokenizes data, resulting in a matrix containing word counts. This matrix is then transformed by the TF-IDF transformer, which weights terms based on its frequency and rarity across data. The joined matrix that results gives a high visual of text data, combining both term frequencies

| Comments Before Preprocessing | Translated Comments Before Preprocessing | Comments After Preprocessing | Translated Comments After Preprocessing |
|---|---|---|---|
| হয়তো ❤ আয়মান ভাইয়ের পেইজের এডমিন $% মুনজেরিন আপু। আইডি সুইচ করতে ভুলে গিয়েছেন। | Maybe ❤ Admin of Ayman Bhai's page $% Munzerin Apu. Forgot to switch ID. | আয়মান ভাইয়ের পেইজের এডমিন মুনজেরিন আপু আইডি সুইচ ভুলে গিয়েছেন | Admin of Ayman Bhai's page Miss Mungerin has forgotten the ID switch." |
| এতো :) আমাদের জন্য বড় # আনন্দের সংবাদ | So much :) Big #happy news for us | আনন্দের এতো বড় সংবাদ | Such great news of joy |
| >এজন্যই বলি আ-লীগে ❤ চলে আসুন। সব নির্বাচন-ই সুষ্ঠু নির্বাচন। | >That's why I say A-League ❤ Come on. All Elections are fair elections. | আলীগে আসুন। এজন্যই নির্বাচন। নির্বাচনই বলি সুষ্ঠু | Come on A-league. That's what elections are for. I call the election fair. |
| এটিএম কার্ড দেখিয়ে স্টুডেন্ট # হাফপাস নেওয়া আমি | Taking Student # Half Pass with ATM :) card i" | এটিএম কার্ড দেখিয়ে স্টুডেন্ট হাফপাস | Student Halfpass by showing ATM card |
| > আমাদের জীবনে কোন সিজি নাই? | > No CG in our lives? | কোম জীবনে সিজি | CG in which life |
| এগিয়ে যাক ❤ মানবতা! | Let's move forward ❤ Humanity! | এগিয়ে মানবতা যাক | Let humanity go forward |
| মহান ~ আল্লাহ রাব্বুল আলামীন ❤ আপনাদের এই সেবা কবুল করুক আমিন | Great ~ Allah Rabbul Alamin ❤ This is yours May the service be accepted Amen | আপনাদের আমিন আলামীন আল্লাহ কবুল করুক মহান রাক্বুল সেবা | May Allah accept your Amin Alamin Servant of God Almighty |
| আপনাদের জন্য সব সময় দোয়া রইলো।মহান আল্লাহ আপনাদের উত্তম প্রতিদান দিবেন ইনশায় আল্লাহ | Praying for you always. Great God God will reward you well. | আপনাদের আল্লাহ ইনশায় উত্তম দিবেন দোয়া প্রতিদান রইলো।মহান সময় | May Allah bless you with good prayers Reward. Great time." |
| $% এগুলা ই বাক্স বন্ধি ভালোবাসা (❤)! | $% These are the box-locked love (❤)! | এগুলা বন্ধি বাক্স ভালোবাসা | These are closed box love |
| দয়া করে কৈফিয়ত দিবেন না কারন আপনাদের উদ্দেশ্যে মহান। ফি আমানিল্লাহ। | Please don't make excuses because of you Great for the purpose. Fee Amanillah. | আপনাদের আমানিল্লাহ উদ্দেশ্যে কারন কৈফিয়ত দিবেন দয়া ফি মহান | Amenillah to you because of the excuse Give kindness fee is great |

**Fig 6. Original and processed text of Bengali text dataset.**

| Sample Dataset: | | |
|---|---|---|
| **Comment** | | **Label** |
| আপনাদের এই ভালো কাজেই রয়েছে জাতির কল্যাণ <br><br> (The welfare of the nation lies in this good work of yours) | | Not Bully |
| আল্লাহ তায়ালা সফল করুক (May Allah make it successful) | | Religious |
| হাসি ফুটিয়ে যান এভাবেই (Smile like this) | | Not Bully |

**Token Indices from Count Vectorization:**

['আপনাদের' 'আল্লাহ' 'এভাবেই' 'করুক' 'কল্যাণ' 'কাজেই' 'জাতির' 'তায়ালা' 'ফুটিয়ে' 'ভালো' 'রয়েছে' 'সফল' 'হাসি']

['may" 'Allah' 'so" 'do' 'wellness' 'so" 'nation' 'tayala' 'bloom' 'good' 'remain' 'success' 'smile']

**Count Vectorization Output:**

[[1 0 0 0 1 1 1 0 0 1 1 0 0] [0 1 0 1 0 0 0 1 0 0 0 1 0] [0 0 1 0 0 0 0 0 1 0 0 0 1]]

**Token Indices from TF-IDF Transformation:**

['X0' 'X1' 'X2' 'X3' 'X4' 'X5' 'X6' 'X7' 'X8' 'X9' 'X10' 'X11' 'X12']

**TF-IDF Transformation Output:**

[[0.40824829 0.       0.       0.       0.40824829 0.40824829 0.40824829 0.       0.       0.40824829 0.40824829 0. 0.]
 [0.       0.5       0.       0.5       0.       0. 0.       0.5       0.       0.       0.       0.5 0.]
 [0.       0.       0.57735027 0.       0.       0. 0.       0.       0.57735027 0.       0.       0. 0.57735027]]

**Fig 7. Example of CV & TF-IDF transformation representation of a comment.**

and importance. This hybrid model improves ML classifiers for tasks such as cyberbullying analysis and classification, as well as NLP applications by giving a better understanding of character importance in documents. "Fig 7" shows us an example of CV and TF-IDF transformation representations of a sample dataset, and the translated version of the Bangla text is in brackets.

The following Eq (1) is used to visualize the explained technique: Where *Tf* refers to the TF-IDF transformation employed to the matrix gated from CV, *Cv* refers to the use of CV to tokenize and count words, resulting in a matrix, and *Hy* is the final matrix that paired both term frequencies and significance on TF-IDF weighting.

$$Hy = Tf\left(Cv\left(data\right)\right) \tag{1}$$

**Tokenization and padding for deep learning models.** The use of tokens serves as a vital assignment when analyzing text in raw form over cyberbullying assignments. Tokenization is a method of separating written material through tokens of value, that are fragments regarding the initial content. We tokenized the document during each word stage as well as allocated symbols for every word. The dataset includes phrases of varied lengths, which can impact the accuracy of the categorization techniques. As a result, we used the edging methodology to standardize the measurement for every phrase. In this procedure, an extra equal is utilized during short phrases and words that go over the maximum allowed length are diminished [29]. "Fig 8" shows us an example of tokenization and padding representation of sample comment, and the translated version of the comment is in brackets below.

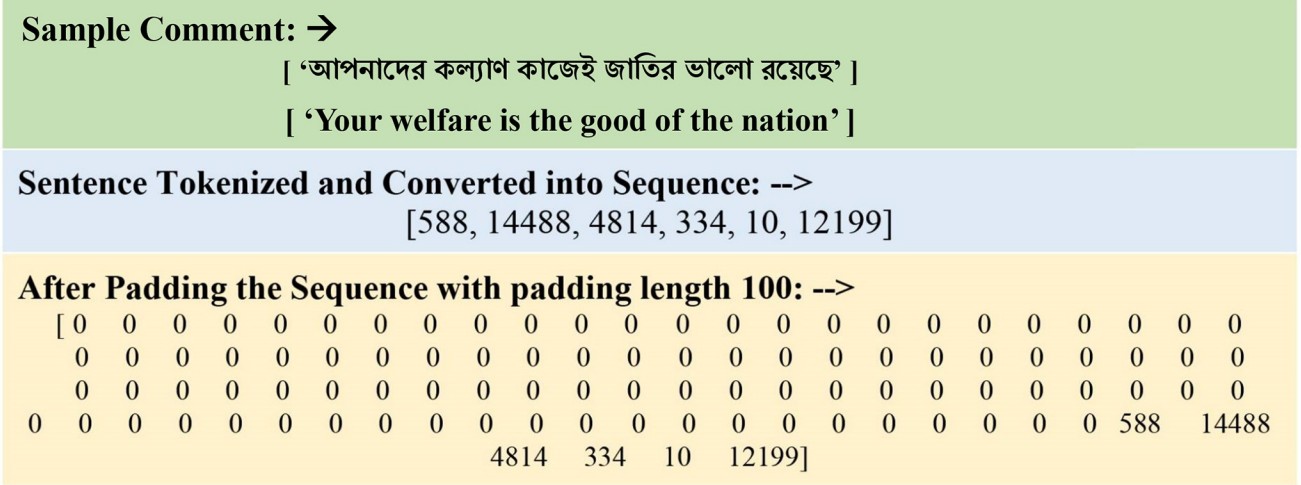

**Fig 8. Example of tokenization and padding representation of a comment.**

### Resampling the data

The process of reprocessing entails frequently extracting specimens of the original data [30]. We used the underestimating approach because we had a lot of textual information, which allowed us to create a harmonious dataset that best reflects reality and can detect Bengali bullying. This method of leveling uneven data entails retaining each of the information that comes to the minority class and reducing the percentage of the majority class. To extract more accurate data, this method makes use of initially disorganized text in the collection process. To minimize the dominant group and reduce imbalances in class, the underestimating is done by employing a new method called Instance Hardness Threshold (IHT) [31]. As for the algorithm's learning, we used a logistic regression algorithm in IHT and cross-validation. Logistic regression is easier to employ, understand, as well as to train.

### Machine learning classifiers

In order to create a framework to recognize Bengali bullying, this research measures performance using ten classifiers, including Staking, Voting, SGD, LR, RF, MLP, Bagging, XGBoost, Adaboost, and K-NN. A variety of performance indicators have been used to evaluate each classifier's performance. We will discuss the different ML techniques used in this study for the model of prediction and categorization strategies that are outlined in this section.

**Stacking ensemble.** To improve the accuracy of predictions, ensemble training is a mixed ML method that considers the forecasting capabilities of several basic algorithms [32]. Three categories of methods are used in combination development: bagging, boosting, as well as stacking. In this particular research, several ML algorithms are employed initially within a single-level arranging system. Lastly, finalized estimates are returned by fitting a LogitBoost to the forecasts from each categorization framework. "Fig 9" shows the arrangement of the stacking group.

**Voting classifier.** A voting classification is a method of collaborative learning that employs several separate classification algorithms and aggregates its forecasts to potentially outperform one algorithm [33]. The opportunity to predict the class designation instead of the final class mark which is frequently predicted using description models is provided by hard

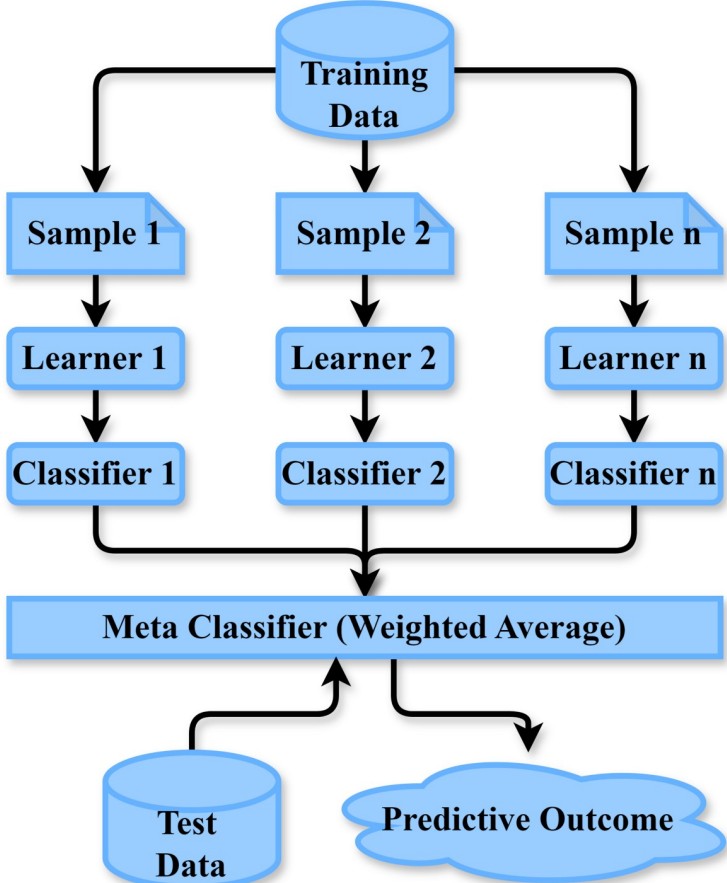

**Fig 9. Arrangement of the stacking ensemble.**

voting. By an average of the class possibilities, scholars can predict what classes will be called using soft voting [34]. "Fig 10" shows us an arrangement of the voting ensemble model.

**Stochastic gradient descent.**   The goal of achieving a task that has the correct excellence characteristics can be advanced repeatedly using the SGD method [35]. It determines the level of progress based on the evolution of additional factors. Because it substitutes an indicator over the actual angle which is calculated via the entire educational index for the angle which is identified using an arbitrary portion of the data, it is quite accurate and could be seen to provide an unpredictable estimate of tendency decrease improvement [36]. Rather than precisely calculating the gradient of $G_n(j_s)$, every time it predicts this value using an only determined for instance $k_r$:

$$s_{r+1} = s_r - \gamma_r \nabla_s P\left(k_r, s_r\right) \tag{2}$$

**Logistic regression.**   LR objective is to use the evaluation and training information collections as a foundation for classifying the comments through multiple orientation classifications. When used on new data, it performs admirably [37]. Its formula in numbers is:

$$K^{\beta 0 + \beta 1 k 1 + \beta 2 k 2} \tag{3}$$

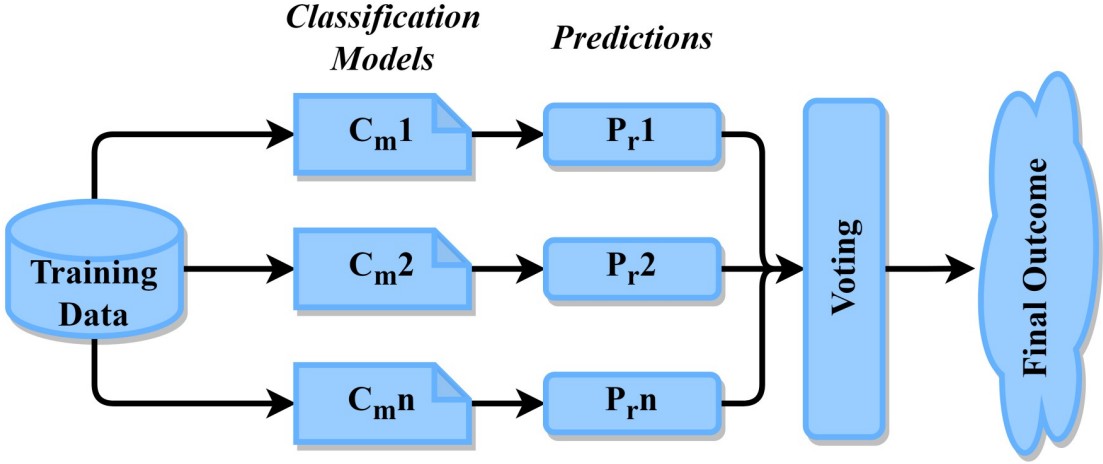

**Fig 10. Arrangement of the voting ensemble.**

Where *K* denotes a dependent value and the amount that remains represents the expression's boundaries operation.

**Multi-layer perceptron.** In each set of inputs, a neural network known as the MLP creates a unique vector in order to function. Included in the MLP multilayered perceptron structure are an informational stage, an output, and a hidden stage [38]. Eq (4) represents the logical modification of the initial information by every single layer of the Multi-Layer Process (MLP), implementing the stimulation operation that every single layer as well as utilizing weights and biases, as defined by Pacifici et al. [37]. The vector that is being input is *S*, the bias vector is $m^{(i)}$, the value of the weight matrix for layer *i* is denoted by $T^{(i)}$, and the activated operation is represented by $\Psi$.

$$K = \Psi\left(T^{(n)}\Psi\left(T^{(n-i)}\ldots\Psi\left(T^{(i)}S + m^{(i)}\right) + m^{(n-i)}\right) + m^{(n)}\right) \tag{4}$$

**K-nearest neighbor.** This method is employed for task reversion and organization. Be aware that certain models, like the one in [39], represent the data set's vectors using the class level. The process of classifying a newly arrived content according to its characteristic vector $p_i$ and its resemblance with the K-NN characteristic vectors *pj* which is computed by applying the geographic vector similarities measure is represented by the formula (5). The integer operates the equation making sure that every single neighbor's contribution is just taken into account in the event that $p_i$ doesn't exist in the total number *Zm*.

$$X(pi, Zm) = \sum_{j=1}^{C} \frac{\sum_{z=1}^{N} EizEij}{\sqrt{\sum_{C=1}^{N} EiC^2}\sqrt{\sum_{C=1}^{N} EjC^2}} \delta(pi, Zm) \tag{5}$$

$$\delta(pi, Zm) = \{0, \textbf{if } pi \in zm1, \textbf{ if } pi \in zm\} \tag{6}$$

**Random Forest.** A "random forest" is a DT configuration where branches are created during the preprocessing stage [40]. This algorithm combines predictions from multiple models in an ensemble approach. The RF will select that class from among all the forest trees based on which classification garnered the most votes, should that class receive any votes. The

mathematical formula for the RF classification algorithm appears as follows:

$$U_{ij} = k_{ij}V_j - k_{left(j)}V_{left(j)} - k_{right(j)}V_{right(j)} \tag{7}$$

**Bagging classifier.** Considering it was developed in 1996 by Leo Breiman, bagging is also known as bootstrap aggregating [41]. In the situation of division into multiple potential categories $X$: $Z \rightarrow \{-1, 1\}$ with respect to the structure of a particular definition $Z$ a collection of training. The sequence of classification algorithms $X_n$, $n = 1 \ldots, N$ is generated by the collecting approach in reaction to modifications in the creating information. Eq (8) displays the mixed classification predictions as the weighted average of the separate classification projections. In order to boost the influence of more precise classification algorithms on the result's estimation, the variables $k_n$, $n = 1 \ldots, N$ are selected.

$$X(zi) = sin\left(\sum\nolimits_{n=1}^{N} k_n X_n(z_i)\right) \tag{8}$$

**Extreme Gradient Boosting.** XGBoost, a higher-level tree approach, applies the gradient-boosting idea [42]. XGBoost employs a more regularized model formalization than previous gradient-boosting techniques to prevent data overfitting [43]. The XGBoost classifier's final projection is represented by $S(r)$ in Eq (9). $P$ is the total quantity of trees or enhancing phases. The phase dimensions or speed of learning at each iteration p is denoted by $\Gamma p$. The $p$-th insufficient classification (tree) in the team is denoted by $hp(r)$.

$$S(r) = \sum\nolimits_{p=1}^{P} \Gamma_p b_p(r) \tag{9}$$

**Adaptive boosting.** This technique is sometimes referred to as weak learning. Weak learning was established in 2017 [44] by guaranteeing the lowest value for each feature's standard threshold value. The weakest classification algorithm is represented by $t_i(r)$ along with $s_i$, which stands for the weighted and weakest classification algorithm beginning at stage $i$, accordingly, in Eq (10).

$$T(r) = \sum\nolimits_{i=1}^{I} s_i t_i(r) \tag{10}$$

## Deep learning models

In order to create a framework to recognize Bengali bullying, this research measures performance using 5 DL models also, including BiGRU, CNN, CLSTM, DNN, and RNN. A variety of performance indicators have been used to evaluate each model's performance. We will discuss the different DL techniques used in this study for the model of prediction and categorization strategies that are outlined in this section.

**Deep neural network.** An improved model designed for multimodal learning is called a DNN. Its complex and well-structured hierarchy highlights this model's adaptability and demonstrates its usefulness in a variety of applications [45]. In essence, the DNN is a cutting-edge approach that builds on blended learning research.

**Recurrent neural network.** A type of neural network, also known as recurrent neural networks [46], has chain reactions because the relationships among their brain cells form an instructed phase. The main function of RNNs is to understand data in batches utilizing the

internal recall that the instructed cycles have recorded. $h_t$ is the hidden state at time $t$, $x_t$ is the input at time $t$, $W_{hh}$ represents the recurrent weights, $W_{xh}$ represents the input weights, $b_h$ is the bias term, $f$ is the activation function (commonly sigmoid or *tanh*).

$$h_t = f(W_{hh}\, h_{t-1} + W_{xh}\, x_t + b_h) \tag{11}$$

**Convolutional neural network.** It was developed to address 2D problems. Among the many that make it up are Conv2D, maximal pooling, flattening, and an entirely linked layer [47]. In Eq 12, $y$ stands for the outcome, $W$ for filter coefficients (kernels), $X$ for the input information, $b$ for biases, $*$ for the kernel convolution procedure, and $f$ for the activated function.

$$y = f(W * X + b) \tag{12}$$

**Convolutional Long Short-Term Memory.** This simulation is similar to what was previously demonstrated. After the CNN had extracted characteristics, this implementation used an LSTM layer to handle decreased excellent input patterns [48]. A CLSTM network's convolutional neural networks as well as recurrent operations are combined in Eq 13. The hidden state at time $t$ is represented by $h_t$, the input at time $t$ is by $x_t$, the input-to-hidden convolutional weights are represented by $W_{ih}$, the hidden-to-hidden recurrent weights are represented by $U_{hh}$, the bias term is by $b_h$, the convolution operation is denoted by $*$, and the activation function (usually sigmoid or *tanh*) is by $f$.

$$h_t = f(W_{ih} * x_t + U_{hh} * h_t - 1 + b_h) \tag{13}$$

**Bidirectional Gated Recurrent Unit.** As a way to regulate the movement of textual data at various points, The GRU paired applicant secret states as well as modified an LSTM's three entry points by restoring as well as modifying entrances [49]. Intelligent prejudice results from the GRU's exclusive focus on past events as well as disregard for the impact of the following data on current words [50]. As a result, a bidirectional simulation, BiGRU, was employed in this study.

## Experimental results and discussions

The findings, an explanation of the recommended strategy, and the creative design will all be covered in this section. The classification results of 10 ML algorithms Staking, Voting, SGD, LR, RF, MLP, Bagging, XGBoost, Adaboost, and K-NN as well as 5 DL models including BiGRU, CNN, CLSTM, DNN, and RNN that are discussed in this section of the paper. We assessed each classifier separately and found the confusion matrix and ROC curve for each model to find the best fit for our dataset.

### Experimental setup

The ML and DL algorithms, an origin-based tool that mixes Python with NLP, were utilized to develop proposed method for detecting online harassment architecture. The model is trained and evaluated using a local computer carried out with text editor Visual Studio Code, NumPy (1.26.1), Matplotlib (3.8.0), Scikit-Learn (1.3.2), Pandas (2.1.2), Keras (2.15.0), TensorFlow (2.15.0), and PyTorch (2.1.1) on a Windows 10 64-bit computer. The computer has 16 GB of RAM, a Ryzen 9 5900HS by AMD x64 processor with 3.30 GHz with Radeon Graphics, and a GPU called NVIDIA GeForce RTX 3050ti with 4 GB of GPU memory.

**Table 3. ANOVA and chi$^2$ tests results.**

| Test Name | P-Value | F-Value | Chi$^2$-Value |
|-----------|---------|---------|---------------|
| ANOVA | 0.0 | 20.99 | - - |
| Chi$^2$ | 1.0 | - - | 825999.76 |

## ANOVA and Chi$^2$ tests

The ANOVA test has a p-value of 0.0. Considering the null assumption is true, the p-value is the likelihood of receiving findings from the test that are no less than the same extreme as those recorded throughout the investigation as shown as "Table 3". A p-value of 0.0 in this instance suggests that the null theory is strongly refuted by the proof. The ANOVA test's F-value, also known as the F-statistic, is roughly 20.99. The proportion to the overall variability of each group's means how much variance inside the groups is measured by the F-value. Greater variation between the group means in relation to the statistical variability throughout the groups is indicated by a higher F-value. The Chi$^2$ test has a p-value of 1.0. In a similar vein, if the null assumption is correct, the p-value here represents the likelihood of receiving test results that are not less than as extreme as those noted during the test. There may not be an important distinction among the noticed and expected frequencies, according to a p-value of 1.0. The Chi$^2$ value is roughly 825999.76. The difference between the actual and anticipated frequencies in the scenario table is measured by the Chi$^2$ statistic, also known as the Chi$^2$ value. The discrepancy between the expected and observed frequencies is quantified. In conclusion, the ANOVA test indicates a substantial variance between the averages of the groups, whereas the Chi-square test shows no statistically significant variance between what was seen and expected frequencies.

## Parameters used

The most important variables associated with the various classifiers employed by NLP algorithms are presented in the following "Table 4", which offers a thorough comparison of them.

## Performance metrics

Several statistical measures, such as the F1 score, precision, recall, accuracy, and sensitivity, are employed to assess the efficiency of the suggested framework. There are four categories of qualities, according to the confusion matrix's conclusion: true positives (TP), true negatives (TN), false positives (FP), and false negatives (FN). When "TP" is used, true instants have been effectively identified. The letter "TN" denotes a result where the recommended approach correctly identified the fruit variety that was incorrectly classified. "FP" refers to the situation where a positive detection was mistakenly identified by the suggested framework. "FN" denotes the situation in which the suggested framework miscalculated the kind of negative detection. Eqs 14–17 compute these quantities [51].

$$Accuracy = \frac{TP + TN}{TP + TN + FP + FN} \text{ x } 100\% \tag{14}$$

$$Precision = \frac{TP}{TP + FP} \text{ x } 100\% \tag{15}$$

**Table 4. Comparative overview of all DL and ML algorithms parameters in used in this study.**

| | Classifier | Parameters and Values | |
|---|---|---|---|
| **Machine Learning Classifiers** | **LR** | penalty: 'l2', C: -1.0, solver: 'lbfgs', multi_class: 'auto' | |
| | **XGBoost** | learning_rate: 0.1, n_estimators: 100, max_depth: 3, min_child_weight: 1, subsample: 1.0, colsample_bytree: 1.0, gamma: 0, reg_alpha: 0, reg_lambda: 1 | |
| | **AdaBoost** | max_depth: 1, n_estimators: 50, learning_rate: 1.0, algorithm: 'SAMME.R', random_state: None | |
| | **RF** | n_estimators: 100, criterion: 'gini', max_depth: None, min_samples_split: 2, min_samples_leaf: 1, min_weight_fraction_leaf: 0, max_leaf_nodes: None, random_state: 42 | |
| | **K-NN** | n_neighbors: 5, weights: 'uniform', algorithm: 'auto', leaf_size: 30, p: 2, metric: 'minkowski' | |
| | **Bagging** | max_depth: 5, n_estimators: 10, max_samples: 1.0, max_features: 1.0, bootstrap: True, random_state: 42, verbose: 0 | |
| | **MLP** | hidden_layer_sizes: 100, activation: 'relu', solver: 'adam', alpha: 0.0001, learning_rate_init: 0.001, max_iter: 200, shuffle: True, random_state: None | |
| | **SGD** | alpha: 0.0001, loss: 'hinge', max_iter: 1000, tol: 0.001, random_state: 42, epsilon: 0.1, penalty: 'l2', learning_rate: 'optimal', verbose: 0 | |
| | **Stacking** | base_estimator: {('sgd', SGDClassifier()), ('mlp', MLPClassifier()), ('rf', RandomForestClassifier())}, final_estimator: RandomForestClassifier(), stack_method: 'auto', n_jobs: -1 | |
| | **Voting** | estimators: [('sgd', SGDClassifier()), ('lr', LogisticRegression()), ('mlp', MLPClassifier())], voting: 'soft' | |
| **Deep Learning Models** | **CNN** | filters: 200, kernel_size: 3, activation: 'relu', Dense layers: 50 units with 'relu' activation, 5 units with 'softmax' activation, optimizer: 'adam', epochs: 15, batch_size: 32, workers: 8 | Vocab Size = 20000, embedding Dim = 300, max_length = 100. |
| | **CLSTM** | filters: 64, kernel_size: 3, activation: 'relu', pool_size: 2, LSTM units: 100, Dense layers: 50 units (ReLU), 5 units (Softmax), learning_rate: 0.0005, epsilon: 1e-07, optimizer: Adam, epochs: 15, batch_size: 32, use_multiprocessing: True, workers: 8 | |
| | **RNN** | SimpleRNN units: 100, Dense layers: 50 units (ReLU), 5 units (Softmax), learning_rate: 0.0005, epsilon: 1e-07, optimizer: Adam, epochs: 15, batch_size: 32, use_multiprocessing: True, workers: 8 | |
| | **DNN** | Dense layers: 256 units (ReLU), 128 units (ReLU), 64 units (ReLU), 5 units (Softmax), learning_rate: 0.0005, epsilon: 1e-07, optimizer: Adam, epochs: 15, batch_size: 32, use_multiprocessing: True, workers: 8 | |
| | **BiGRU** | Bidirectional GRU: 100 units, Return sequences: True (2 layers), Conv1D: Filters = 64, kernel_size = 3, activation: 'relu', MaxPooling1D: pool_size = 2, Dense layers: 128 units (ReLU), 64 units (ReLU), 5 units (Softmax), learning_rate: 0.0005, epsilon: 1e-07, optimizer: Adam, epochs: 10, batch_size: 32, use_multiprocessing: True, workers: 8 | |

$$F_{1score} = 2 \times \frac{Pr \times Re}{Pr + Re} \text{ x } 100\% \tag{16}$$

$$Recall = \frac{TP}{TP + FN} \text{ x } 100\% \tag{17}$$

## Results analysis

We have evaluated the technique we proposed for binary as well as multi-label categorization and discover that the approach achieves notably higher detection outcomes to feed Bengali bullying. Production of Several-Class Categorizing Inside the evaluated multiple categories of the dataset, identically proposed ML as well as DL techniques have been employed under the method of learning different comment data to locate and categorize several kinds of internet-based assault groups (e.g., religious background, troll, threat, sexual, as well as non-bullying). Multiple kind of supervised methods of classification were utilized, and the hypotheses were assessed using outcomes involving accuracy, precision, recall, and the F1 score. The previously impressive performance demonstrated by the hybrid stacking model, which demonstrates a 99.34% accuracy during a classification of several class assignments, highlights the versatility of our methodology. The framework successfully classifies 3,753 comments, and the reliability,

precision, recall, and F1 score assessment indicators all confirm the efficiency of our approach. The careful examination is graphically represented as shown in "Fig 11(a)", in which the confusion matrix shows the small difference of just twenty-five wrongly categorized feedbacks throughout the massive data set. Particularly, as "Fig 11(b)–11(d)" illustrates, LR, MLP, RF, SGD, Voting, and XGBoost stand out just alongside exceptional precision, while Adaboost, Bagging, and K-NN accomplish quietly via a reduced level of accuracy as well as a greater rate of statistical erroneous.

Moving on to binary categorization, our combination stacking model demonstrates how valuable it is once again. With just 85 cases of incorrect categorization, it skillfully classifies an astounding 13,346 remarks, demonstrating the flexibility as well as efficacy of our system when working across a larger information set. Its actives an accuracy of 99.41%, precision of 99.41%, recall of 99.41% along with F1 score of 99.41%. "Fig 12(a)" presents a detailed illustration of the equivalent matrix of confusion for the classification of binary data, demonstrating the system's capacity to handle the subtleties of the form of binary labeling. In addition to the conventional measurements, our research includes a thorough examination of the ROC curve, which is shown in "Fig 12(b)". This curve provides detailed insights into the model's ability to handle a variety of accuracy and specificity by illuminating its performance across a range of limitations. The assessment is deepened by the ROC curve, which enables an enhanced understanding of each model's efficacy attributes.

In this work, we use Receiver Operating Characteristic (ROC) curves to graphically depict the multilevel categorization accomplishment percentage, which serves as an essential component in assessing the effectiveness of different classification methods. ROC curves play a crucial role in demonstrating the compromise within specificity (1-Specificity) as well as sensitivity (True Positive Rate, or TPR), providing insightful information about a classifier's capacity to differentiate among multiple categories. Our study aims to illustrate the ROC curve to feed various classifiers in the specific scenario of multiclass categorization by employing a wide variety of threshold values. The ROC curve normally starts at (0,0), which denotes a situation in which every detail is categorized as negative, as well as ends at (1,1), which acts as a threshold for positive classification. The shape of this curve represents the trade-off between correctly classifying adverse circumstances as beneficial and incorrectly classifying instances that are positive as positive. The multiclass ROC curve for each classifier Adaboost, Bagging, K-NN, LR, MLP, RF, SGD, Voting, XGBoost, and our stacking model is depicted in "Fig 13", which is underneath and also "Table 5" shows the AUC score of all ML models. Every "Fig 13 (a)–13(j)" represents a distinct classifier and displays its own ROC curve. A sophisticated comprehension of every classifier's efficiency across a range of value thresholds is made possible by this thorough analysis. When evaluating the efficacy of a classifier, an ROC curve is a useful tool, especially when dealing with multilabel category examples. One important measurement obtained using the curve of the ROC is known as the Area Under the Curve (AUC), where a greater AUC typically denotes better accuracy of the model. A thorough summary of the way every algorithm and hybrid ML model handles the complexities of distinguishing within each of the five categories taken into consideration in our research is given in "Fig 13".

"Table 5" shows the AUC scores for multiple machine learning models over various kinds of detection. Each model executed extremely well for identifying non-bully, troll, sexual, religious, and threat content, via AUC scores near or equal to 1.00. However, AdaBoost along with K-NN performed slightly worse in some classes, with AUC values spanning 0.59 to 0.99. Overall, the results show that the models are highly effective, with the majority accomplishing near-perfect AUC scores across all classes.

A wide range of outcome measurements, including important metrics like F1 score, recall, accuracy, and precision, are summarized in "Table 6". Simultaneously, "Fig 14" presents the

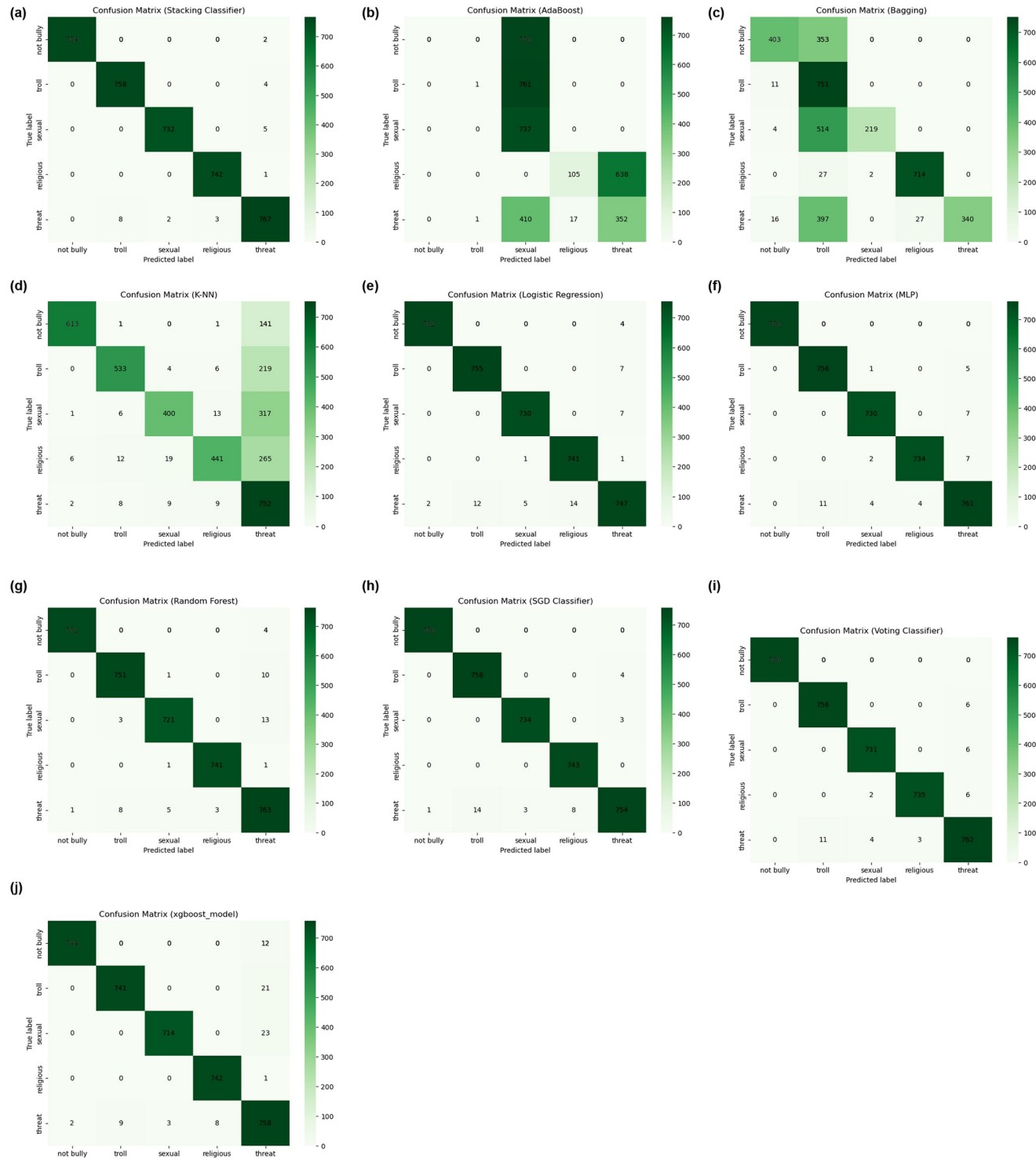

**Fig 11. Results of training as well as validation: The matrix of confusion of all ML models.** (a) Stacking, (b) AdaBoost, (c) Bagging, (d) K-NN, (e) LR, (f) MLP, (g) RF, (h) SGD, (i) Voting, (j) XGBoost.

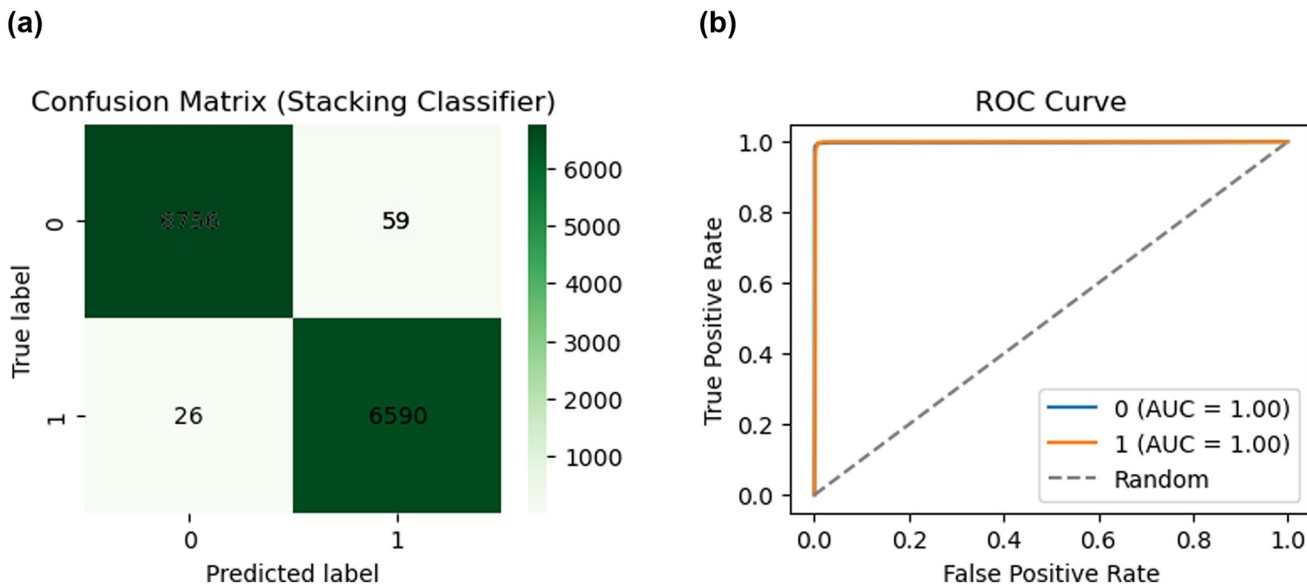

**Fig 12. Confusion matrix and ROC curve of hybrid stacking model with binary classifications.** (a) Confusion Matrix, (b) ROC curve.

identical information in a chart with bars as an illustration. We focus on ML models and analyze their efficiency using these important metrics. Regarding precision, recall, and F1 scores, both the stacking and SGD algorithms perform exceptionally well, with notably high precision percentages of 99.34% and 99.13%, respectively, along with similarly high recalls and F1 scores. The aforementioned models demonstrate their effectiveness in making precise forecasts with a general precision of 99.34% and 99.13%, respectively. The voting and MLP algorithms function as well, through F1 scores reflecting such remarkable results as recall rates of 98.99% and 98.91%, in addition to precision rates of 99% and 98.92%. The corresponding accuracy values are 98.99% and 98.91%, respectively, which demonstrate how consistently accurate they can be. Conversely, Adaboost exhibits smaller recall (31.63%) and precision (39.75%), resulting in an F1 score of 21.49%. Similar to this, the performance of the Bagging and K-NN models lags, with accuracy levels of 64.24% to 72.50%, respectively. The XGBoost, LR, and RF models perform in a balanced manner, achieving accuracy levels above 97%. These frameworks show excellent precision, recall, and F1 score values, confirming their dependability in task categorization circumstances even though they do not achieve the precision specifications for stacking and SGD. Recall that the F1 score and precision are important because they provide a detailed assessment of a model's effectiveness. Recall highlights the model's capacity to remember all real positive tests; precision measures how accurate the framework is at predicting optimistic situations; and the F1 score finds a balance among the two to provide a thorough evaluation of an algorithm's overall performance. Here, stacking and SGD remain as algorithms that demonstrate their proficiency in handling the assignment at hand by being able to both predict positives precisely and record everything genuine positives in a timely way. Overall, "Table 6" along with "Fig 14" clearly illustrates the privilege of careful assessment, which highlights the specific advantages and disadvantages of every ML framework. It also gives us a clear idea of how every approach performs across important metrics and confirms that stacking as well as SGD are effective in the aforementioned multilabel categorization instance.

We use multiple DL simulations, which include DNN, CNN, BiGRU, RNN, and CLSTM, to analyze various comment sets of data as well as classify various types of internet attacks. We

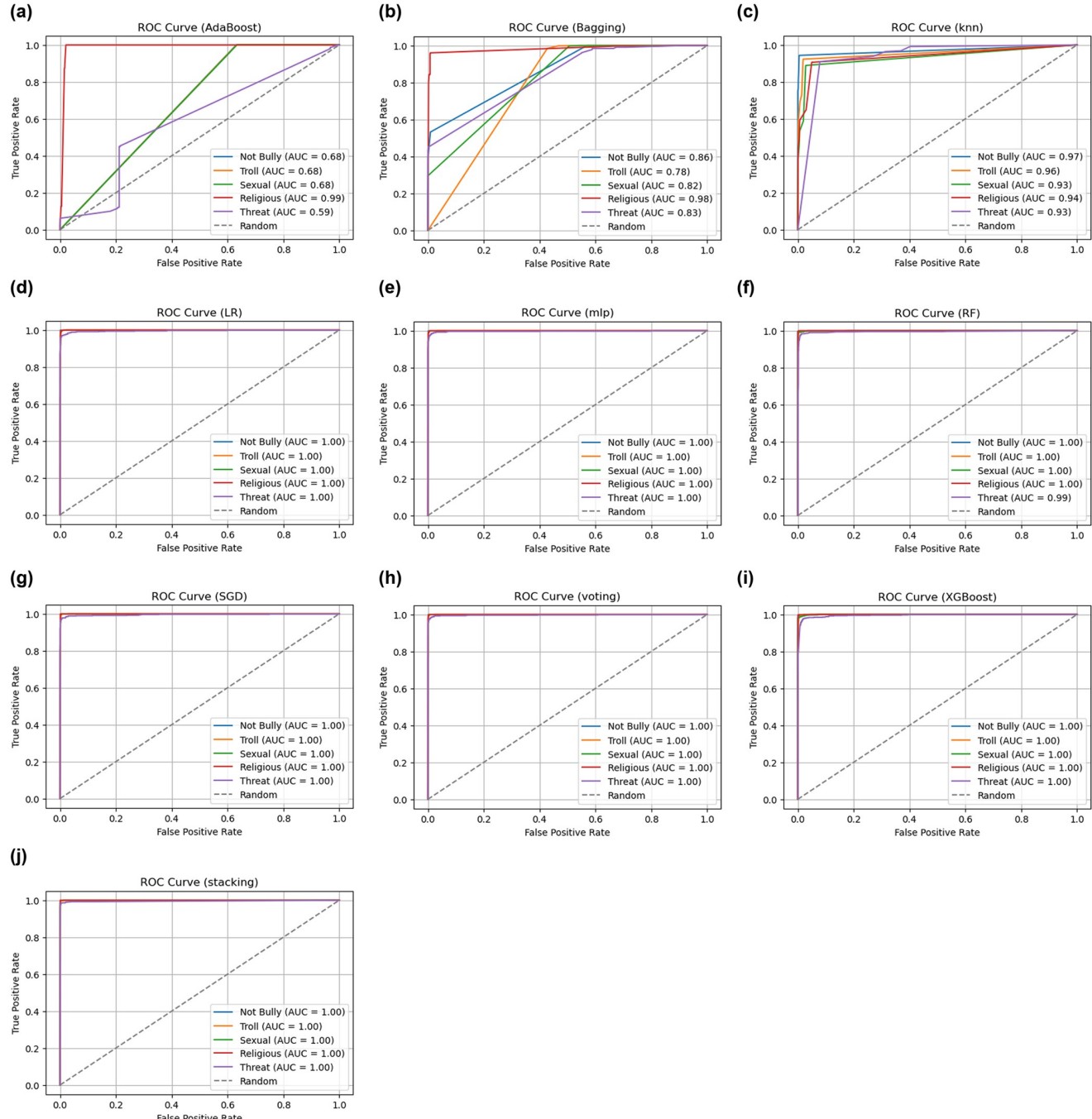

**Fig 13. All ML models ROC curve over every class.** (a) Adaboost, (b) Bagging, (c) K-NN, (d) LR, (e) MLP, (f) RF, (g) SGD, (h) Voting, (i) XGBoost, (j) Stacking.

assess the predictions using a variety of overseen classification techniques and major performance indicators, which include accuracy, precision, recall, and the F1 score. Interestingly, our DL algorithms execute exceptionally well. The previous BiGRU approach, for example, reached an excellent accuracy of 90.63% when classifying various classes. This demonstrates how flexible and effective our approach is at managing different types of cyberbullying. The

**Table 5. All ML models AUC score list.**

| Classifiers | AUC score | | | | |
|---|---|---|---|---|---|
| | Not Bully | Troll | Sexual | Religious | Threat |
| LR | 1.00 | 1.00 | 1.00 | 1.00 | 1.00 |
| XGBoost | 1.00 | 1.00 | 1.00 | 1.00 | 1.00 |
| AdaBoost | 0.68 | 0.68 | 0.68 | 0.99 | 0.59 |
| RF | 1.00 | 1.00 | 1.00 | 1.00 | 0.99 |
| K-NN | 0.97 | 0.96 | 0.93 | 0.94 | 0.93 |
| Bagging | 0.86 | 0.78 | 0.82 | 0.98 | 0.83 |
| MLP | 1.00 | 1.00 | 1.00 | 1.00 | 1.00 |
| SGD | 1.00 | 1.00 | 1.00 | 1.00 | 1.00 |
| Stacking | 1.00 | 1.00 | 1.00 | 1.00 | 1.00 |
| Voting | 1.00 | 1.00 | 1.00 | 1.00 | 1.00 |

**Table 6. All ML models classification reports of precision, recall, F1 score, and accuracy.**

| Classifiers | Accuracy | Precision | Recall | F1 Score |
|---|---|---|---|---|
| Stacking | 99.34% | 99.34% | 99.34% | 99.34% |
| Voting | 98.99% | 99.00% | 98.99% | 98.99% |
| SGD | 99.13% | 99.13% | 99.13% | 99.12% |
| LR | 98.60% | 98.60% | 98.60% | 98.59% |
| MLP | 98.91% | 98.92% | 98.91% | 98.91% |
| K-NN | 72.50% | 84.60% | 72.50% | 74.27% |
| RF | 98.68% | 98.68% | 98.68% | 98.68% |
| Bagging | 64.24% | 84.93% | 64.24% | 64.74% |
| XGBoost | 97.91% | 98.01% | 97.92% | 97.95% |
| AdaBoost | 31.63% | 39.75% | 31.63% | 21.49% |

**Fig 14. The collection of data assessment's accuracy, precision, recall, and F1 score of all ML models.**

comprehensive analysis is illustrated graphically in "Fig 15(a)", in which the confusion matrix reveals a small disparity, using 354 comments not correctly categorized throughout the huge amount of data. The dependability of our method has been confirmed by a thorough evaluation of accuracy, precision, and recall, along with the F1 score, for which the structure skillfully classifies 3,424 feedback comments. Notable is the contribution of various DL algorithms, namely BiGRU, CLSTM, CNN, DNN, and RNN, which contribute to the general efficacy of our methodology, as depicted in "Fig 15(a)–15(e)" in that order. This demonstrates how well predictive models handle the complexities of multi-label grouping for identifying harassment in Bengali. Our investigation of DL models confirms their importance in the field of online misconduct identification by demonstrating their capacity to both supplement and, in some cases, outperform conventional ML methods. The technique we use has been modified to utilize DL models, which broadens its application and offers a flexible or all-encompassing solution to the complex problem of classifying and identifying different types of internet assault in Bengali language information.

The "Fig 16" and "Table 7" below, shows the multiclass ROC curve and AUC score for each model, which includes BiGRU, CLSTM, CNN, DNN, and RNN. For each classifier shown in "Fig 16(a)–16(e)", a unique ROC curve is displayed. This comprehensive analysis allows a sophisticated understanding of each classifier's efficiency across a range of value thresholds. "Fig 16" provides an extensive summary of how each model addresses the challenges of differentiating within each of the five categories that our study examined.

"Table 7" compares the AUC scores of multiple deep learning models that identify different types of content. BiGRU, CNN, CLSTM, DNN, along with RNN all had high AUC ratings, indicating superior performance across classes. While the AUC scores differ slightly between models and classes, they consistently outperform in detecting non-bully, troll, sexual, religious, or threat content.

DL models analyze them using some important metrics. Regarding precision, recall, and F1 scores, both the BiGRU and CNN algorithms perform exceptionally well, with a notably significant precision rate of 90.73% and 90.03%, respectively, along with similarly high recalls and F1 scores. The CLSTM, DNN, and RNN algorithms function as well, through F1 scores reflecting such remarkable results as recall rates of 89.36%, 89.20%, and 88.41%, in addition to precision rates of 89.59%, 89.57, and 88.37%. The corresponding accuracy values are 89.36%, 89.20%, and 88.41%, respectively, which demonstrate how consistently accurate they can be. These frameworks show excellent precision, recall, and F1 score values, confirming their dependability in task categorization circumstances even though they do not achieve the precision specifications for BiGRU and CNN. Overall, "Table 8" along with "Fig 17" clearly illustrates the privilege of careful assessment, which highlights the specific advantages and disadvantages of every DL framework. It also gives us a clear idea of how every approach performs across important metrics and confirms that BiGRU and CNN are effective in the aforementioned multilabel categorization instance.

## Discussion

The dissertation presents an in-depth examination of all the specifics regarding our hybrid ML strategy, which is specifically designed to identify instances of online bullying in Bengali-language written material, using the findings from the experiment as well as subsequent debates. As the authors of the paper, we provide a thorough examination of a wide range of ML as well as DL scenarios, illuminating both the particular benefits of each one and the overall effectiveness of the structure that was created. Ten algorithms namely, Staking, Voting, SGD, LR, RF, MLP, Bagging, XGBoost, and Adaboost along with K-NN are examined in detail concerning

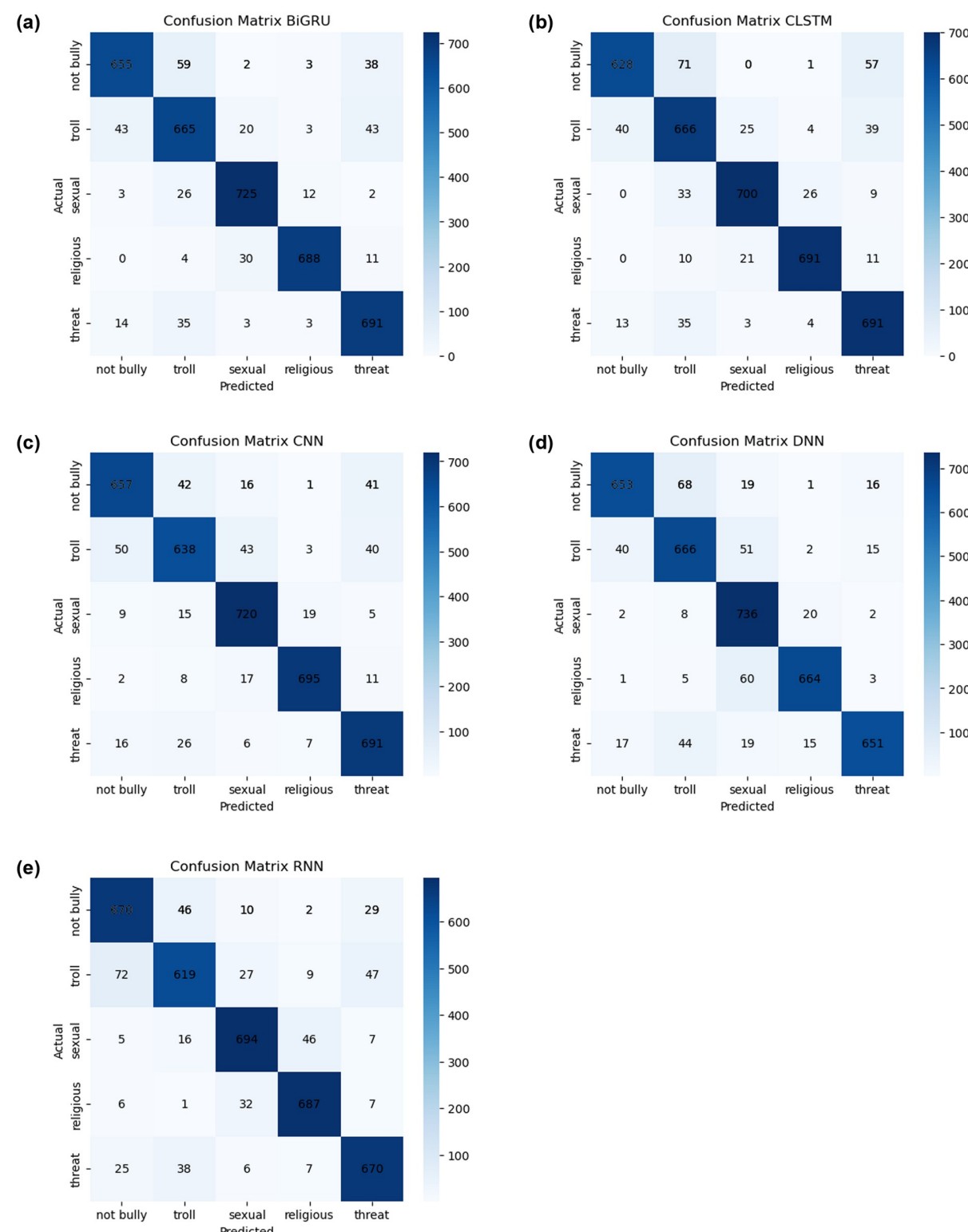

**Fig 15. Results of training as well as validation: The matrix of confusion of all ML models.** (a) BiGRU, (b) CLSTM, (c) CNN, (d) DNN, (e) RNN.

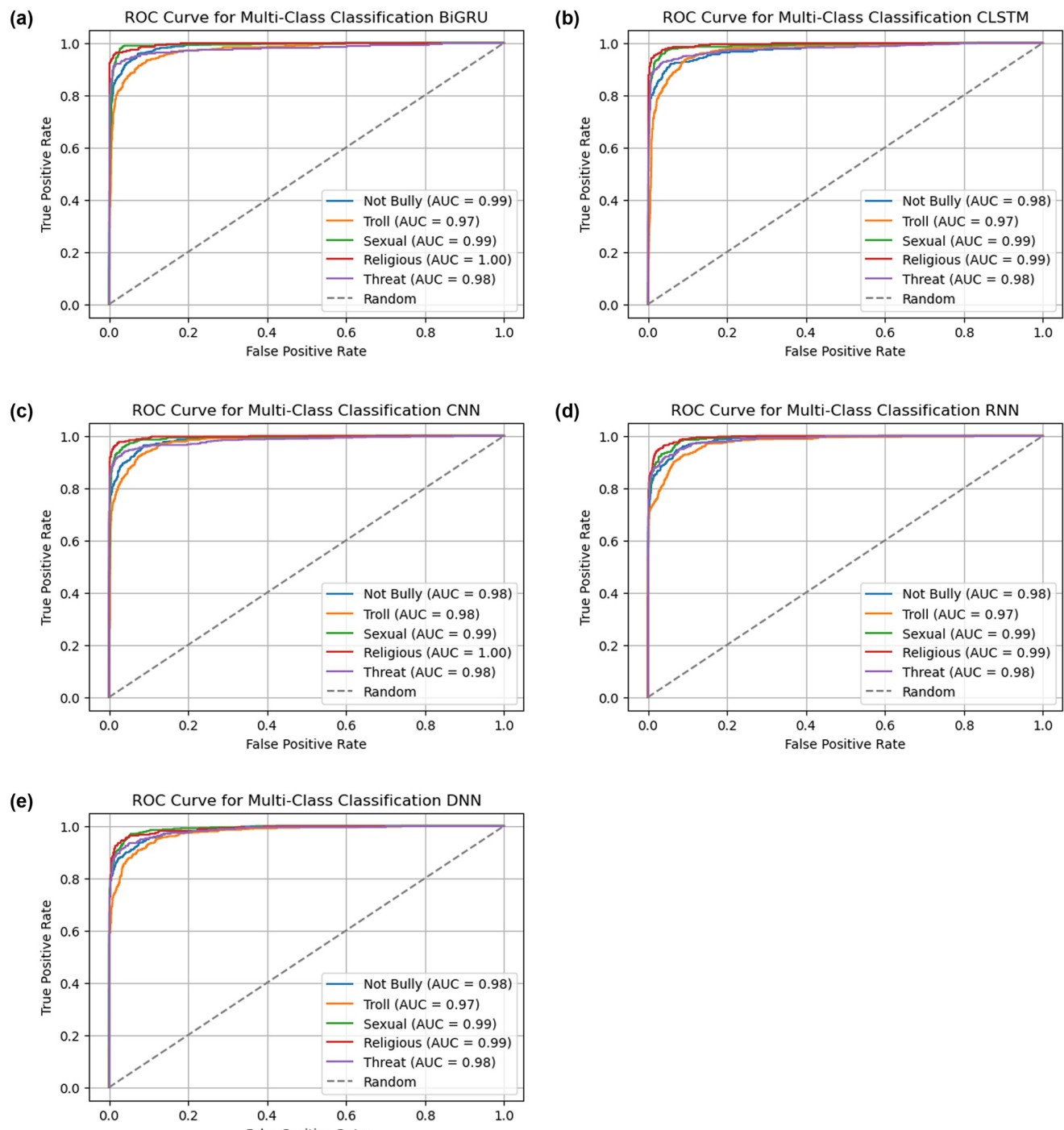

**Fig 16. All DL models ROC curve over every class.** (a) BiGRU, (b) CLSTM, (c) CNN, (d) RNN, (e) DNN.

binary as well as multi-label classifications as an element of the inspection of ML models. The best performer is the hybrid stacking model, which achieves an amazing accuracy of 99.34% through multi-class classification. The algorithm's remarkable accuracy and recall, along with its F1 score in an assortment of internet abuse classifications, including sexually explicit

**Table 7. AUC score comparison table for all the deep learning algorithms.**

| Classifiers | AUC score | | | | |
|---|---|---|---|---|---|
| | Not Bully | Troll | Sexual | Religious | Threat |
| BiGRU | 0.99 | 0.97 | 0.99 | 1.00 | 0.98 |
| CNN | 0.98 | 0.98 | 0.99 | 1.00 | 0.98 |
| CLSTM | 0.98 | 0.97 | 0.99 | 0.99 | 0.98 |
| DNN | 0.98 | 0.97 | 0.99 | 0.99 | 0.98 |
| RNN | 0.98 | 0.97 | 0.99 | 0.99 | 0.98 |

**Table 8. All DL models classification report of precision, recall, F1 score and accuracy.**

| Classifiers | Accuracy | Precision | Recall | F1 Score |
|---|---|---|---|---|
| BiGRU | 90.63% | 90.73% | 90.63% | 90.64% |
| CNN | 90.02% | 90.03% | 90.02% | 89.98% |
| CLSTM | 89.36% | 89.59% | 89.36% | 89.38% |
| DNN | 89.20% | 89.57% | 89.20% | 89.24% |
| RNN | 88.41% | 88.37% | 88.41% | 88.36% |

material, threats, trolls, faith histories, and non-bullying incidents, highlight its adaptability along with its general effectiveness. A more thorough grasp of the pros and cons of each ML simulation is made possible by the careful examination of confusion matrices alongside ROC curves. Models with notable precision include LR, MLP, RF, SGD, and Voting, as well as XGBoost. Models with lower accuracy and a higher rate of statistical errors include Adaboost, Bagging, and K-NN. This complicated analysis helps identify which approaches are the best suited to tackle the complex issues that harassment online in Bengali raises. The research investigation highlights the potential of DL algorithms, among them BiGRU, CNN, CLSTM, DNN, and RNN, for the classification of various forms of cyberbullying within Bengali text.

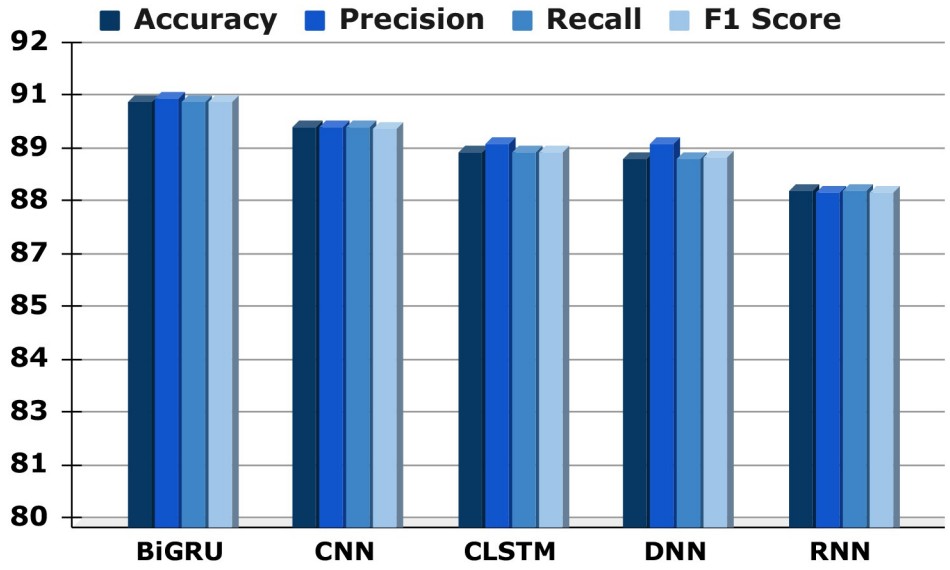

**Fig 17. The collection of data assessment's accuracy, precision, recall, and F1 score.**

With an exceptional accuracy rate of 90.63%, the BiGRU model stands out, specifically highlighting the effectiveness of DL methods in addressing the convoluted nature of internet assault. Precision, recall, and F1 scores are also used to evaluate DL frameworks. BiGRU, along with CNN, both demonstrate noteworthy precision rates. This adds to the increasing amount of evidence that supports the utility of DL frameworks in the field of internet assault identification by reaffirming their capabilities.

A thorough examination of past research suggests that our research is a trailblazer in the field of NLP identification methods. The current study has a collection of 94,000 instances, which is significantly larger than the typical dataset dimensions in similar studies. Managing a demanding five-class multi-class issue, our investigation shows a sophisticated comprehension of various language subtleties. The biggest surprise is the remarkable performance of 99.34%, which attests to the system's effectiveness in properly identifying instances in every group. This excellent accuracy highlights both the suggested system's resilience and its opportunity for practical use. Furthermore, every category frequently achieves a 99.34% precision, recall, and F1 score, indicating a harmonious simulation that excels in obtaining an excellent proportion of pertinent data in addition to adequacy. "Fig 18" shows the classification report of hybrid stacking model.

The research's supremacy is reinforced by a comparison alongside the additional research in "Table 9". Haque et al. [5] utilized various models including SVC, SGD, RF, LR, MNB, DT, CLSTM, BiGRU, BiLSTM, and LSTM, achieving an accuracy of 85.80% with a dataset size of 42,036. Their limitation was lower detection accuracy. Das et al. [11] employed RNN, attention mechanism, LSTM, GRU, and CNN, achieving an accuracy of 77% with a dataset size of 7,425. They faced challenges with lower detection accuracy and limited dataset coverage. Eshan et al.

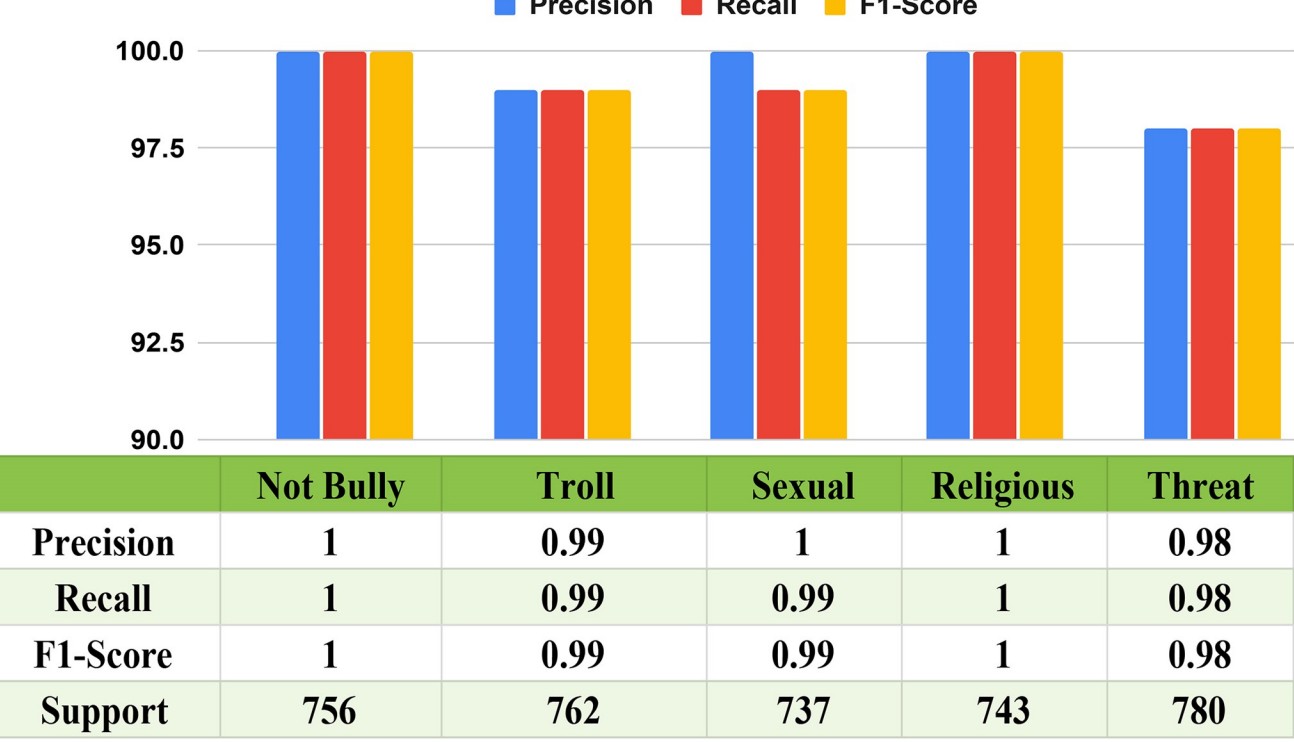

|  | Not Bully | Troll | Sexual | Religious | Threat |
|---|---|---|---|---|---|
| Precision | 1 | 0.99 | 1 | 1 | 0.98 |
| Recall | 1 | 0.99 | 0.99 | 1 | 0.98 |
| F1-Score | 1 | 0.99 | 0.99 | 1 | 0.98 |
| Support | 756 | 762 | 737 | 743 | 780 |

**Fig 18. Classification report of proposed hybrid model.**

Table 9. Comparison of the proposed model with previous investigations.

| References | Feature Extraction Technique | | | Dataset Size | Classes | Accu-racy | Precession | Recall | F1 Score | AUC score |
|---|---|---|---|---|---|---|---|---|---|---|
| Haque et al. [5] | TF-IDF | Count Vectorizer | Word Embedding | 42,036 | 4 | 85.80 | 86.00 | 85.00 | 86.00 | - - |
| Das et al. [11] | TF-IDF | - - | Word Embedding | 7,425 | 7 | 77 | - - | - - | - - | - - |
| Eshan et al. [12] | - - | - - | - - | 2,500 | - - | 75 | - - | - - | - - | - - |
| Ishmam et al. [13] | - - | - - | Word2Vec Embedding | 5,126 | 6 | 70.10 | 68.00 | 70.00 | 69.00 | - - |
| Ahmed et al. [14] | - - | - - | Word Embedding & Tokenization | 44,001 | 2 | 87.91 | 90.00 | 75.00 | 82.00 | - - |
| | | | | | 5 | 85.00 | 85.00 | 85.00 | 84.00 | - - |
| Ahmed et al. [15] | TF-IDF | - - | Word Embedding | 12,000 | 2 | 80.00 | 80.00 | 80.00 | 80.00 | 0.50 |
| Emon et al. [16] | TF-IDF | - - | Word Embedding | 4,700 | 7 | 82.20 | 83.00 | 82.00 | 82.00 | - - |
| Mahmud et al. [17] | TF-IDF | | - - | 3,000 | 2 | 97.00 | 95.00 | 94.00 | 95.00 | - - |
| M. Khan et al. [18] | TF-IDF | | - - | 63,000 | 7 | 62.00 | - - | - - | 58.00 | - - |
| Romim et al. [19] | TF-IDF | - - | Word Embedding | 50,314 | 2 | - - | - - | - - | 86.78 | - - |
| Akhter et al. [20] | TF-IDF | - - | - - | 2,400 | - - | 97.73 | 98.00 | - - | 98.00 | 0.54 |
| Akhter et al. [21] | TF-IDF | - - | - - | 44,001 | 5 | 98.82 | 98.86 | 98.89 | 98.88 | - - |
| | | | | | 2 | 98.57 | 98.58 | 98.56 | 98.56 | 0.997 |
| Our Work | TF-IDF | Count Vectorizer | Tokenization | 94,000 | 5 | 99.34 | 99.34 | 99.34 | 99.34 | - - |
| | | | | | 2 | 99.41 | 99.41 | 99.41 | 99.41 | 1.00 |

[12] used RF, MNB, and SVM, achieving an accuracy of 75% with a dataset size of 2,500. They also faced limitations in accuracy and dataset coverage. Ishmam et al. [13] employed GRU, Adaboost, RF, NV, and SVC, achieving an accuracy of 70.10% with a dataset size of 5,126. They encountered limitations in accuracy and dataset coverage. Ahmed et al. [14] applied Random Forest, SVM, KNN, Naïve Bayes, and a hybrid neural network (CNN-SVM), achieving an accuracy of 87.91% for CNN-SVM and 85% for SVM with a dataset size of 44,001. Their limitations included being limited to specific categories and having a dataset with limited accuracy. Ahmed et al. [15] utilized MNB, SVM, LR, XGBoost, CNN, LSTM, BLSTM, and GRU, achieving an accuracy of 80% for MNB with a dataset size of 12,000. They faced limitations in accuracy and dataset coverage, and also achieved an AUC score of 0.80. Emon et al. [16] employed LinearSVC, LR, MNB, RF, ANN, RNN, and LSTM, achieving an accuracy of 82.20% with a dataset size of 4,700. They faced limitations due to a relatively small dataset and lower detection accuracy. Mahmud et al. [17] utilized LR, MB, DT, RF, SVM, AdaBoost, GB, SGD, ET, KNN, and MLP, achieving an accuracy of 97% with a dataset size of 3,000. They faced limitations due to a relatively small dataset and a limited preprocessing approach. M. Khan et al. [18] employed SVM, NB, RF, KNN, and NN, achieving an accuracy of 62% with a dataset size of 63,000. Their limitations included lower detection accuracy and only 2% of data labeled for the "Religious" class. Romim et al. [19] utilized Bi-LSTM, achieving an F1-score of 86.78% with a dataset size of 50,314. Their study was limited to hate speech detection in Bengali language, and they did not discuss the potential impact of the dataset. Akhter et al. [20] utilized NB, J48, SVM, and KNN, achieving an accuracy of 97.73% with a dataset size of 2,400, and also achieved an AUC score of 0.54. They faced limitations in accuracy and dataset coverage. Akhter et al. [21] employed DT, RF, LR, and MLP, achieving an accuracy of 98.82% for MLP and 98.57% for LR with a dataset size of 44,001, and also achieved an AUC score of 0.997. They did not use DL models, and their models' execution time was high.

Using a 94,000 dataset, our work used TF-IDF with Count Vectorizer and Tokenization, resulting in an accuracy of 99.34% for the 5-class classification task and 99.41% for the 2-class

classification task, and also achieved an AUC score of 1.0. In terms of accuracy, our method performs better than current approaches. Compared with other studies, we achieve significantly higher accuracy by using TF-IDF with Count Vectorizer and Tokenization. Furthermore, our approach tackles the drawbacks noted in earlier research, including reduced detection precision, restricted dataset scope, and particular linguistic limitations.

Most importantly, we incorporate both Count Vectorizer with TF-IDF transformer and Tokenization in its feature extraction technique, showcasing a thoughtful integration of methods for enhanced performance. The versatility and effectiveness of the model are enhanced by this all-encompassing approach. The Bengali Cyberbullying Detector (BCBD), a web-based application built using the hybrid stacking approach (SGD+MLP+LR), is a prime instance of a real-life application. We have become a leader in the domain of natural language processing (NLP)-based detection procedures by providing an attractive blend of a large dataset, skillful multi-class categorization, and a strong feature extraction method.

## Building web application

In accordance with the suggested hybrid approach (SGD+MLP+LR) developed through stacking, we developed an online tool designated Bengali Cyberbullying Detector (BCBD) for performing multi-class categorization upon Bengali texts employing the Flask platform. This tool is designed for chancing multi-class categorization in Bengali text. By classifying online bullying into five different orders, BCBD tries to empower individuals to depict the true of online bullying within Bengali text. The significance of BCBD lies in its capability to help druggies decode and comprehend the factual online bullying environment present in a given text. This categorization allows druggies to respond quickly to the content without any confusion. "Fig 19" shows a visual representation of BCBD, demonstrating how druggies input their target text. The bracket results are also displayed, indicating the linked order for each input. The successful integration of the SGD, MLP, and LR factors in our approach enhances the delicacy and effectiveness of BCBD. This tool not only contributes to the field of natural language processing but also addresses the pivotal issue of cyberbullying, particularly in the Bengali environment.

## Conclusion and future work

The research emphasizes multi-class online bullying about Bengali comments on social networking sites, an issue complicated by little study. focusing on the difficulties of identification to be a result of perpetrators obscuring away from conflict. A hybrid ML approach to detecting Bengali online bullying is suggested, including text preprocessing, a combination of feature extraction techniques for ML classifiers as well as tokenization over DL models, and data managing using IHT. ML techniques (Staking, Voting, SGD, LR, RF, MLP, Bagging, XGBoost, Adaboost, and K-NN) along with DL models (BiGRU, CNN, CLSTM, DNN, and RNN) utilized with k-fold cross-validation technique, while efficiency is measured through different indicators. We categorized 94,000 Bengali online comments through five different groups using data gathered from two datasets that are open to the publicduring the research, we demonstrated that multi-class cyberbullying was able to attain an exceptional level of correctness over the language by using successful processing and extraction of features using different methodologies. The algorithm accurately detected Bengali online harassment on social media platforms with a success rate of about 99.41% as well as 99.34% in binary along with multiclass categorization, correspondingly, by employing a hybrid ML approach, as well as 90.63% correctness employing a DL algorithm named BiGRU. According to our findings, the model given might prove beneficial to computerized Bengali cyberbullying surveillance systems.

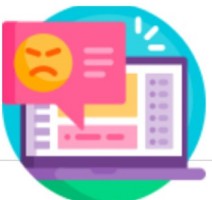

Bangla Cyberbullying Detector (BCBD)
# Cyberbullying Detection Using Artificial Intelligence

## Cyberbullying Detection From Bangla Text

## The Cyberbullying Type is: Religion

নাস্তিক অমুসলিম বেদ্বিন বেঈমান, আল্লাহ আপনি আমাদের সবাইকে হিফাজ্ত করুন ইমান আমল নিয়ে জীবন যাপন করার তৌফিক দীন আপনি আমাদের সবাইকে সহীহ বোঝ দীন আমিন

**Predict**

**Fig 19. Web view of our system.**

There are a few restrictions to the present investigation. For starters, its primary emphasis on detecting online Bengali cyberbullying through a hybrid ML technique is limited to specific languages as well as cultural environments. The application of an openly accessible data set containing few comments might not accurately reflect the wide variety of cyberbullying incidents. The investigation fails to examine transformer-based methods, that showed potential in tasks involving natural language processing. Furthermore, difficulties and concerns of practical application are not investigated. Managing these drawbacks would improve the dependability as well as the relevance of future identification of cyberbullying studies.

Though the outcomes is good, there is always an opportunity for enhancement. In future years, we will conduct studies to tackle the problem of interspersed classes by broadening the information on training. Our team would like to investigate transformer-based techniques. Methods including BERT, RoBERTa, and ELECTRA, among others might have applications on multilingual online bullying databases covering a variety of cyberbullying classifications. This field of investigation has the potential to improve the effectiveness of online harassment surveillance systems. On the opposite end of the spectrum, we are researching video and picture filtration because both serve as targets for harassment on social networking sites.

## Author Contributions

**Conceptualization:** Khandaker Mohammad Mohi Uddin, Md Ashraf Uddin.

**Data curation:** Hasibul Hamim, Mst. Nishat Tasnim Mim, Arnisha Akhter.

**Formal analysis:** Hasibul Hamim, Arnisha Akhter.

**Investigation:** Hasibul Hamim, Arnisha Akhter.

**Methodology:** Khandaker Mohammad Mohi Uddin, Hasibul Hamim.

**Resources:** Khandaker Mohammad Mohi Uddin.

**Software:** Khandaker Mohammad Mohi Uddin, Hasibul Hamim.

**Supervision:** Khandaker Mohammad Mohi Uddin, Md Ashraf Uddin.

**Validation:** Hasibul Hamim, Mst. Nishat Tasnim Mim.

**Visualization:** Hasibul Hamim.

**Writing – original draft:** Khandaker Mohammad Mohi Uddin, Mst. Nishat Tasnim Mim, Md Ashraf Uddin.

**Writing – review & editing:** Arnisha Akhter, Md Ashraf Uddin.

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
