## [Decision Letter · Decision Letter 0]

11 Mar 2024

PONE-D-24-03168Machine Learning and Deep Learning-Based Hybrid Strategy to Categorize Bengali Comments on Social Networks Using a Fused DatasetPLOS ONE

Dear Dr. Mohi Uddin,

Thank you for submitting your manuscript to PLOS ONE. After careful consideration, we feel that it has merit but does not fully meet PLOS ONE’s publication criteria as it currently stands. Therefore, we invite you to submit a revised version of the manuscript that addresses the points raised during the review process.

We look forward to receiving your revised manuscript.

Kind regards,

Muhammad Bilal

Academic Editor

PLOS ONE

Reviewers' comments:

Reviewer's Responses to Questions

**Comments to the Author**

1. Is the manuscript technically sound, and do the data support the conclusions?

Reviewer #1: Yes

Reviewer #2: Yes

2. Has the statistical analysis been performed appropriately and rigorously? 

Reviewer #1: Yes

Reviewer #2: No

3. Have the authors made all data underlying the findings in their manuscript fully available?

Reviewer #1: Yes

Reviewer #2: Yes

4. Is the manuscript presented in an intelligible fashion and written in standard English?

Reviewer #1: Yes

Reviewer #2: Yes

5. Review Comments to the Author

Reviewer #1: 1. The title should be reviewed, especially the word "hybrid strategy" and "a" should be removed before the words "Fused Dataset.

2. The authors used so many acronyms first without given providing the full meaning such as CNN, LSTM, TF-IDF, NN, ML, DL, etc. Make sure that these acronyms are defined first, then you can use the acronym later.

3. Some Literature Reviews (LR) are not based on Bengali Language, I think, these LR should be removed.

4. A comparison table for the deep learning algorithms used in the study should be provided for the ROC CURVE Matric as well.

5. In the discussion, three (3) LR were only compared with your findings, what about the remaining LR? It should be compared as well.

Reviewer #2: The paper discusses the impact of social media on personal issues and the need to address online harassment. The study uses machine learning and natural language processing to analyze Bengali comments for harassment detection, employing various models such as MLP, K-NN, XGBoost, etc., and DL frameworks like DNN, CNN, C-LSTM, and BiLSTM. The hybrid model (MLP+SGD+LR) showed the best performance with 99.34% accuracy, precision, and F1 score for multi-label classification and 99.41% accuracy for binary classification. However, here are some suggestions to improve the quality of the manuscript.

1. Enhance Figure Descriptions: Add more detailed captions to figures to explain their relevance and findings more clearly.

2· Clarification of Technical Terms: Provide a glossary or a more detailed explanation of technical terms and acronyms for readers not familiar with the field.

3. Consistency in Formatting: Ensure consistency in formatting throughout the paper, including figure labels, table headings, and references, to enhance readability.

4. Expand Discussion Section: Offer a more comprehensive comparison with existing methods, including advantages and potential limitations of the proposed approach.

5. It would be beneficial to include statistical tests in the manuscript to provide a more robust analysis of the results. Utilizing tests such as ANOVA, t-tests, or chi-square tests can help to validate the significance of the findings and strengthen the overall credibility of the study.

6. Extend the literature by adding new and related to consumer electronics citations.

7. Code Availability: Consider adding a section on the availability of code or data used in the study to enhance transparency and reproducibility.

6. PLOS authors have the option to publish the peer review history of their article (what does this mean?). If published, this will include your full peer review and any attached files.

Reviewer #1: No

Reviewer #2: No

---

## [Author Response · Author response to Decision Letter 0]

22 May 2024

Manuscript Number: PONE-D-24-03168

Title: Machine Learning and Deep Learning-Based Hybrid Strategy to Categorize Bengali Comments on Social Networks Using a Fused Dataset.

Journal: PLOS ONE.

Dear Editor, 

Thank you for allowing a resubmission of our manuscript, with an opportunity to address the reviewers’ comments. Below is the point-by-point response to the comments, changes are made according to the reviewers' comments and a revised version of the manuscript is attached. All modifications to the paper are highlighted using RED Color. 

Response to reviewer’s comments: 

Reviewer #1: 

Concern 1. The title should be reviewed, especially the words "hybrid strategy" and "a" should be removed before the words "Fused Dataset.

Author response:

Thank you for your valuable feedback. We appreciate your thorough review of our manuscript. We will revise the title by removing the words "hybrid strategy" and "a" before "Fused Dataset" to enhance its precision and conciseness.

Author Action:

Machine Learning and Deep Learning-Based Approach to Categorize Bengali Comments on Social Networks Using Fused Dataset

Concern 2. The authors used so many acronyms first without given providing the full meaning such as CNN, LSTM, TF-IDF, NN, ML, DL, etc. Make sure that these acronyms are defined first, then you can use the acronym later. 

Author response:

Thank you for your valuable feedback on our manuscript. We have ensured that these acronyms are defined before their subsequent use in the document. We also add a section with the acronyms.

Author Action:

AdaBoost Adaptive Boosting LightGBM Light Gradient-Boosting Machine

ANN Artificial Neural Network MLP Multi-Layer Perceptron

BiGRU Bidirectional Gated Recurrent Unit MNB Multinomial Naive Bayes

BERT Bidirectional Encoder Representations from Transformers NLP Natural Language Processing

Bi-LSTM Bidirectional Long Short-Term Memory RF Random Forest

CLSTM Convolutional Long Short-Term Memory RNN Recurrent Neural Network

CNN Convolutional Neural Network SN Social Networks

DT Decision Tree SA Sentiment Analysis

DNN Deep Neural Network SVM Support Vector Machine

ET Extra Tree SGD Stochastic Gradient Descent

IHT Instance Hardness Threshold TF-IDF Term Frequency-Inverse Document Frequency

KNN K-Nearest Neighbor XGBoost Extreme Gradient Boosting

LR Logistic Regression 

Concern 3. Some Literature Reviews (LR) are not based on the Bengali Language, I think, these LRs should be removed.

Author response:

Thanks for your suggestion. As per your suggestion, to better align our literature review with a focus on the Bengali language, we can consider removing all non-Bengali language literature reviews. However, in doing so, it's essential to ensure that we maintain the depth and breadth of our research. We can aim to replace these reviews with relevant literature in the Bengali language, thus creating a more comprehensive and inclusive review that aligns with our research objectives.

Author Action:

…………... 

However, utilizing an encoder-decoder-based LSTM network, Das et al. [11] proposed a different approach for identifying instances of hateful language in Bengali. To address the various classes hateful language issue, they employed TF-IDF, word2Vec, and a 1D CNN model in a network using LSTM. An encoder-decoder machine learning model a well-liked NLP tool was presented in this paper as a way to categorize feedback submitted by users in Bengali on profiles on Facebook.

Eshan et al. [12] also examined a variety of machine learning algorithms, including Random Forest, multinomial Naïve Bayes, Support Vector Machine with linear, Radial Basis Function (RBF), Polynomial, and Sigmoid kernels. They also contrasted these algorithms with characteristics utilizing single-word, bigram, and trigrams, which are used by Count Vectorizer and Tfidf Vectorizer, but found that the SVM Linear kernel produced the most effective outcomes.

A Gated Recurrent Unit (GRU) model was developed by Ishmam et al. [13] to classify feedback from customers on social media sites. 5126 pieces of Bengali feedback were gathered for the purpose of research, and they were divided into five categories: political discourse, spiritual remarks, encouragement, insults, and discrimination towards race and religion. The GRU structure recognized hateful speech with an accuracy of 70.10%.

………………….

Utilizing machine learning techniques like NB, J48, SVM, as well as KNN, Akhter et al. [20] completed an analogous binary categorization assignment to identify Bengali cyberbullying comments. A collection of 2400 Bengali comments classified as either attacked or not was used in their tests. They used the TF-IDF as an extractor of features to teach the SVM classification algorithm, and they were able to achieve 97% accuracy. Nevertheless, multi-class SA was not present in the experiment because it tended toward categorization into binary categories.

…………..

11. Das, A. K., Al Asif, A., Paul, A., & Hossain, M. N. (2021). Bangla hate speech detection on social media using attention-based recurrent neural network. Journal of Intelligent Systems, 30(1), 578-591.

12. Eshan, S. C., & Hasan, M. S. (2017, December). An application of machine learning to detect abusive bengali text. In 2017 20th International conference of computer and information technology (ICCIT) (pp. 1-6). IEEE.

13. Ishmam, A. M., & Sharmin, S. (2019, December). Hateful speech detection in public facebook pages for the bengali language. In 2019 18th IEEE international conference on machine learning and applications (ICMLA) (pp. 555-560). IEEE.

……..

20. Akhter, S. (2018, December). Social media bullying detection using machine learning on Bangla text. In 2018 10th International Conference on Electrical and Computer Engineering (ICECE) (pp. 385-388). IEEE.

Concern 4. A comparison table for the deep learning algorithms used in the study should be provided for the ROC CURVE Matric as well. 

Author response:

Thank you for your suggestion. We appreciate your emphasis on the importance of providing a comparison table for the deep learning algorithms used in the study, including the ROC curve metric. In response to your concern, we will include a comparison table specifically for the deep learning algorithms, incorporating the ROC curve metric. This addition will provide a more comprehensive comparison of the performance of these algorithms in our study. Thank you for your valuable feedback, which will help enhance the quality of our study.

Author Action:

Table 5 shows the AUC scores for multiple machine learning models over various kinds of detection. Each model executed extremely well for identifying non-bully, troll, sexual, religious, and threat content, via AUC scores near or equal to 1.00. However, AdaBoost along with K-NN performed slightly worse in some classes, with AUC values spanning 0.59 to 0.99. Overall, the results show that the models are highly effective, with the majority accomplishing near-perfect AUC scores across all classes.

Table 5. ML models AUC score list.

Classifiers AUC score

 Not Bully Troll Sexual Religious Threat

LR 1.00 1.00 1.00 1.00 1.00

XGBoost 1.00 1.00 1.00 1.00 1.00

AdaBoost 0.68 0.68 0.68 0.99 0.59

RF 1.00 1.00 1.00 1.00 0.99

K-NN 0.97 0.96 0.93 0.94 0.93

Bagging 0.86 0.78 0.82 0.98 0.83

MLP 1.00 1.00 1.00 1.00 1.00

SGD 1.00 1.00 1.00 1.00 1.00

Stacking 1.00 1.00 1.00 1.00 1.00

Voting 1.00 1.00 1.00 1.00 1.00

Table 7 compares the AUC scores of multiple deep learning models that identify different types of content. BiGRU, CNN, CLSTM, DNN, along with RNN all had high AUC ratings, indicating superior performance across classes. While the AUC scores differ slightly between models and classes, they consistently outperform in detecting non-bully, troll, sexual, religious, or threat content.

Table 7. AUC score comparison table for all the deep learning algorithms.

Classifiers AUC score

 Not Bully Troll Sexual Religious Threat

BiGRU 0.99 0.97 0.99 1.00 0.98

CNN 0.98 0.98 0.99 1.00 0.98

CLSTM 0.98 0.97 0.99 0.99 0.98

DNN 0.98 0.97 0.99 0.99 0.98

RNN 0.98 0.97 0.99 0.99 0.98

Table 9. Comparison of the proposed model with previous investigations

References Feature Extraction Technique Dataset Size Classes Accu-racy Precession Recall F1 Score AUC score

Haque et al. [5] TF-IDF Count Vectorizer Word Embedding 42,036 4 85.80 86.00 85.00 86.00 --

Das et al. [11] TF-IDF -- Word Embedding 7,425 7 77 -- -- -- --

Eshan et al. [12] -- -- -- 2,500 -- 75 -- -- -- --

Ishmam et al. [13] -- -- Word2Vec

Embedding 5,126 6 70.10 68.00 70.00 69.00 --

Ahmed et al. [14] -- -- Word Embedding & Tokenization 44,001 2 87.91 90.00 75.00 82.00 --

 5 85.00 85.00 85.00 84.00 --

Ahmed et al. [15] TF-IDF -- Word Embedding 12,000 2 80.00 80.00 80.00 80.00 0.50

Emon et al. [16] TF-IDF -- Word Embedding 4,700 7 82.20 83.00 82.00 82.00 --

Mahmud et al. [17] TF-IDF -- 3,000 2 97.00 95.00 94.00 95.00 --

M. Khan et al. [18] TF-IDF -- 63,000 7 62.00 -- -- 58.00 --

Romim et al. [19] TF-IDF -- Word Embedding 50,314 2 -- -- -- 86.78 --

Akhter et al. [20] TF-IDF -- -- 2,400 -- 97.73 98.00 -- 98.00 0.54

Akhter et al. [21] TF-IDF -- -- 44,001 5 98.82 98.86 98.89 98.88 --

 2 98.57 98.58 98.56 98.56 0.997

Our Work TF-IDF Count Vectorizer Tokenization 94,000 5 99.34 99.34 99.34 99.34 --

 2 99.41 99.41 99.41 99.41 1.00

Concern 5. In the discussion, three (3) LR were only compared with your findings, what about the remaining LR? It should be compared as well.

Author response:

Thank you for your concern. We acknowledge the oversight in the discussion where only three LR were compared with our findings. We appreciate your attention to detail and apologize for the incomplete comparison. In response, we will update the discussion section to include a comprehensive comparison with all LR mentioned in the study. This will provide a more thorough analysis of our findings in relation to the existing literature. We appreciate your valuable feedback and the opportunity to improve the quality of our study.

Author Action:

Haque et al. [5] utilized various models including SVC, SGD, RF, LR, MNB, DT, CLSTM, BiGRU, BiLSTM, and LSTM, achieving an accuracy of 85.80% with a dataset size of 42,036. Their limitation was lower detection accuracy. Das et al. [11] employed RNN, attention mechanism, LSTM, GRU, and CNN, achieving an accuracy of 77% with a dataset size of 7,425. They faced challenges with lower detection accuracy and limited dataset coverage. Eshan et al. [12] used RF, MNB, and SVM, achieving an accuracy of 75% with a dataset size of 2,500. They also faced limitations in accuracy and dataset coverage. Ishmam et al. [13] employed GRU, Adaboost, RF, NV, and SVC, achieving an accuracy of 70.10% with a dataset size of 5,126. They encountered limitations in accuracy and dataset coverage. Ahmed et al. [14] applied Random Forest, SVM, KNN, Naïve Bayes, and a hybrid neural network (CNN-SVM), achieving an accuracy of 87.91% for CNN-SVM and 85% for SVM with a dataset size of 44,001. Their limitations included being limited to specific categories and having a dataset with limited accuracy. Ahmed et al. [15] utilized MNB, SVM, LR, XGBoost, CNN, LSTM, BLSTM, and GRU, achieving an accuracy of 80% for MNB with a dataset size of 12,000. They faced limitations in accuracy and dataset coverage. Emon et al. [16] employed LinearSVC, LR, MNB, RF, ANN, RNN, and LSTM, achieving an accuracy of 82.20% with a dataset size of 4,700. They faced limitations due to a relatively small dataset and lower detection accuracy. Mahmud et al. [17] utilized LR, MB, DT, RF, SVM, AdaBoost, GB, SGD, ET, KNN, and MLP, achieving an accuracy of 97% with a dataset size of 3,000. They faced limitations due to a relatively small dataset and a limited preprocessing approach. M. Khan et al. [18] employed SVM, NB, RF, KNN, and NN, achieving an accuracy of 62% with a dataset size of 63,000. Their limitations included lower detection accuracy and only 2% of data labeled for the "Religious" class. Romim et al. [19] utilized Bi-LSTM, achieving an F1-score of 86.78% with a dataset size of 50,314. Their study was limited to hate speech detection in Bengali language, and they did not discuss the potential impact of the dataset. Akhter et al. [20] utilized NB, J48, SVM, and KNN, achieving an accuracy of 97.73% with a dataset size of 2,400. They faced limitations in accuracy and dataset coverage. Akhter et al. [21] employed DT, RF, LR, and MLP, achieving an accuracy of 98.82% for MLP and 98.57% for LR with a dataset size of 44,001. They did not use DL models, and their models' execution time was high.

Reviewer #2: 

The paper discusses the impact of social media on personal issues and the need to address online harassment. The study uses machine learning and natural language processing to analyze Bengali comments for harassment detection, employing various models such as MLP, K-NN, XGBoost, etc., and DL frameworks like DNN, CNN, C-LSTM, and BiLSTM. The hybrid model (MLP+SGD+LR) showed the best performance with 99.34% accuracy, precision, and F1 score for multi-label classification and 99.41% accuracy for binary classification. However, here are some suggestions to improve the quality of the manuscript. 

Concern 1. Enhance Figure Descriptions: Add more detailed captions to figures to explain their relevance and findings more clearly. 

Author response:

Thank you for your constructive feedback regarding the figure descriptions. We acknowledge your suggestion to enhance them by adding more detailed captions to explain their relevance and findings more clearly. We will diligently work on improving the clarity and comprehensibility of our figure descriptions to better convey the significance of our findings.

Concern 2. Clarification of Technical Terms: Provide a glossary or a more detailed explanation of technical terms and acronyms for readers not familiar with the field. 

Author response:

Thank you for your valuable suggestion. As per your suggestion, we have revised the manuscript to include a glossary that provides a more detailed explanation of technical terms and acronyms used throughout the paper. This will help readers who may not be familiar with the field to better understand the content. We appreciate your input, and we believe that this addition will enhance the clarity and accessibility of our work.

Author Action:

AdaBoost Adaptive Boosting LightGBM Light Gradient-Boosting Machine

ANN Artificial Neural Network MLP Multi-Layer Perceptron

BiGRU Bidirectional Gated Recurrent Unit MNB Multinomial Naive Bayes

BERT Bidirectional Encoder Representations from Transformers NLP Natural Language Processing

Bi-LSTM Bidirectional Long Short-Term Memory RF Random Forest

CLSTM Convolutional Long Short-Term Memory RNN Recurrent Neural Network

CNN Convolutional Neural Network SN Social Networks

DT Decision Tree SA Sentiment Analysis

DNN Deep Neural Network SVM Support Vector Machine

ET Extra Tree SGD Stochastic Gradient Descent

IHT Instance Hardness Threshold TF-IDF Term Frequency-Inverse Document Frequency

KNN K-Nearest Neighbor XGBoost Extreme Gradient Boosting

LR Logistic Regression 

Concern 3. Consistency in Formatting: Ensure consistency in formatting throughout the paper, including figure labels, table headings, and references, to enhance readability. 

Author response:

Thanks for your suggestion. As per your suggestion, we have carefully reviewed the formatting of our paper and made necessary adjustments to ensure consistency. This includes ensuring uniform formatting for figure labels, table headings, and references throughout the manuscript. We appreciate your attention to detail, and we believe that these improvements will enhance the overall readability of our paper.

Concern 4. Expand Discussion Section: Offer a more comprehensive comparison with existing methods, including advantages and potential limitations of the proposed approach. 

Author response:

Thank you for your valuable suggestion to expand the discussion section. We appreciate your insight and will certainly work on offering a more comprehensive comparison with existing methods. This will i

---

## [Decision Letter · Decision Letter 1]

12 Jul 2024

PONE-D-24-03168R1Machine Learning and Deep Learning-Based Approach to Categorize Bengali Comments on Social Networks Using Fused DatasetPLOS ONE

Dear Dr. Mohi Uddin,

Thank you for submitting your manuscript to PLOS ONE. After careful consideration, we feel that it has merit but does not fully meet PLOS ONE’s publication criteria as it currently stands. Therefore, we invite you to submit a revised version of the manuscript that addresses the points raised during the review process.

We look forward to receiving your revised manuscript.

Kind regards,

Muhammad Bilal

Academic Editor

PLOS ONE

Journal Requirements:

Reviewers' comments:

Reviewer's Responses to Questions

**Comments to the Author**

1. If the authors have adequately addressed your comments raised in a previous round of review and you feel that this manuscript is now acceptable for publication, you may indicate that here to bypass the “Comments to the Author” section, enter your conflict of interest statement in the “Confidential to Editor” section, and submit your "Accept" recommendation.

Reviewer #2: All comments have been addressed

Reviewer #3: (No Response)

Reviewer #4: All comments have been addressed

2. Is the manuscript technically sound, and do the data support the conclusions?

Reviewer #2: Yes

Reviewer #3: Yes

Reviewer #4: Yes

3. Has the statistical analysis been performed appropriately and rigorously? 

Reviewer #2: Yes

Reviewer #3: Yes

Reviewer #4: Yes

4. Have the authors made all data underlying the findings in their manuscript fully available?

Reviewer #2: Yes

Reviewer #3: Yes

Reviewer #4: Yes

5. Is the manuscript presented in an intelligible fashion and written in standard English?

Reviewer #2: Yes

Reviewer #3: Yes

Reviewer #4: No

6. Review Comments to the Author

Reviewer #2: I have examined the revised manuscript submitted by the author, which addresses the comments and concerns raised in the previous review. The author has made substantial improvements and provided clear clarifications in response to the feedback. I strongly recommend the manuscript to be published in the current form.

Reviewer #3: The authors has done a great job, however thera re few suggession to improve the quality of paper. In section 2, related work, each sentence is strated with a reference number, it is suggested to use some mix mode.

2. Add contibution section in the last part of introduction section.

Reviewer #4: The paper has been revised thoroughly; however, there are still a few comments:

Figure 2: Ensure the same font family and size are used consistently across all text elements for uniformity.

Figure 5: Comments before preprocessing should also be translated into English for the reader to understand the preprocessing.

The basic details of the methods, such as Stochastic Gradient Descent, Logistic Regression, Multi-layer Perceptron, Random Forest, Bagging Classifier, Adaptive Boosting, Deep Neural Network, Recurrent Neural Network, Convolutional Neural Network, Long Short-Term Memory, and Bidirectional Gated Recurrent Unit, seem unnecessary and make the paper lengthy. Consider condensing this section to focus on the most relevant methods to streamline the paper. Figures 10 and 11 also seem unnecessary.

7. PLOS authors have the option to publish the peer review history of their article (what does this mean?). If published, this will include your full peer review and any attached files.

Reviewer #2: **Yes: **Farrukh Hassan

Reviewer #3: No

Reviewer #4: No

---

## [Author Response · Author response to Decision Letter 1]

28 Jul 2024

Manuscript Number:PONE-D-24-03168R1

Title:Machine Learning and Deep Learning-Based Hybrid Strategy to Categorize Bengali Comments on Social Networks Using a Fused Dataset.

Journal: PLOS ONE.

Dear Editor, 

Thank you for allowing a resubmission of our manuscript, with an opportunity to address the reviewers’ comments. Below is the point-by-point response to the comments, changes are made according to the reviewers' comments and a revised version of the manuscript is attached. All modifications to the paper are highlighted using RED Color. 

Response to reviewer’s comments: 

Reviewer #2: I have examined the revised manuscript submitted by the author, which addresses the comments and concerns raised in the previous review. The author has made substantial improvements and provided clear clarifications in response to the feedback. I strongly recommend the manuscript to be published in the current form.

Author response:

Thank you very much for your thorough examination of our revised manuscript and your positive feedback. We are delighted to hear that you found our revisions satisfactory and that you recommend our work for publication in its current form. Your constructive comments and suggestions have been invaluable in improving the quality of our manuscript. Your encouraging words mean a lot to us and inspire us to continue striving for excellence in our research. Thanks again for your time and effort in reviewing our work.

Reviewer #3: The authors has done a great job, however there are few suggession to improve the quality of paper.

Author response:

Thank you very much for your positive feedback on our manuscript and for acknowledging our efforts. We are truly grateful for your suggestions to further improve the quality of our paper. Your insights are highly valued, and we will carefully address each point you have raised to enhance our work.We deeply appreciate the time and effort you have invested in reviewing our manuscript and providing us with constructive feedback.

Concern 1.In section 2, related work, each sentence is strated with a reference number, it is suggested to use some mix mode.

Author response:

Thank you for your valuable feedback. We deeply appreciate the time and effort you have taken to review our manuscript. Regarding your concern about Section 2, "Related Work," we have made revisions to address this issue. Instead of starting each sentence with a reference number, we have adopted a mixed mode of citation. This integration should improve the readability and flow of the section. If there are any further adjustments or additional concerns, please let us know. Thank you once again for your valuable suggestions.

Author Action:

………………………………………………………………………………………………………………………………………..A binary and multilabel classifier algorithm was presented by Ahmed et al. [14] to recognize abuse statements on Facebook pages.44,001 user reviews from well-known public Facebook pages were examined for this study and were divided into classes such as non-bully, sexual, threat, troll, and religious. This NN + Ensemble method produced a multilabel classification accuracy of 85% and a binary classification accuracy of 87.91%.

A model for identifying cyberbullying in texts written in Bangla and Romanized was developed by Ahmed et al. [15] using ML and DL techniques.Three social media datasets were produced by their research: one of them for Bangla, another for Romanized Bangla, and one combined dataset. In the combined dataset, the ML algorithm Multinomial Naive Bayes (MNB) achieved an accuracy rate of 80%.

A Bengali-language method for identifying cyberbullying on social media was proposed by Emon et al. [16].They used 44,001 Bengali comments from Facebook to test different transformer models, such as Bengali BERT, Bengali DistilBERT, and XLM-RoBERTa. Out of all the models, the XLM-RoBERTa model had the highest accuracy rate (85%) and F1 score (86%).

A technique for using ML algorithms to recognize abusive language in Bangla was proposed by Mahmud et al. [17].Using logistic regression (LR) and annotated translated Bengali corpora, they were able to identify bullying in Bengali with a 97% accuracy rate.

63,000 Bengali Facebook comments from various celebrity pages were compiled by Khan et al. [18] in order to group fans' sentiments toward the celebrity into five categories: happy, excited, upset, shocked, and content. The feature extractor they used to train SVM, NB, RF, KNN, and NN was TF-IDF. They used the SVM classifier to predict a person's attitude toward a celebrity with a 62% accuracy rate. Even though they employed a sizable dataset for their investigation, the dataset's class imbalance issue resulted in a low accuracy score.

The HS-BAN slanderous speech database in Bengali, which has more than 50,000 categorized statements, was made available by Romim et al. [19]. They investigated language features along with artificial neural network-based methods to develop a common detection of hateful speech systems for Bengali. These comparisons demonstrated that sentences incorporating algorithms developed on unofficial papers performed better than those developed on official texts, leading to the Bi-LSTM models outperforming Fast Text casual word implementation with an F1 score of 86.78%.……………………………………………………………………………………………………………………………….

Concern 2. Add contibution section in the last part of introduction section.

Author response:

Thank you for your valuable feedback. We appreciate your thorough review of our manuscript. Regarding your concern about the contributions section, we have added a new contributions section at the end of the introduction. This section outlines the key contributions of our work to provide readers with a clear understanding of the novel aspects and significance of our research. Thank you once again for your valuable suggestions.

Author Action:

………………………………………………………………..

The main outcomes of the present study paper are as follows:

• A hybrid machine-learning method for detecting online harassment.

• Cleansed data according to several criteria, such as removing punctuation, numerous spaces, particular characters, non-Bengali text, and numbers.

• For the training and testing of the framework, two datasets are combined.

• Count-Vectorizer, along with TF-IDF for ML and tokenization with padding for DL models are used to extract the feature.

• For efficient undersampling, Iterative Hard Thresholding (IHT) was employed. 

• K-Fold cross-validation was used to evaluate the model robustly. 

• Tested for feature significance and model efficacy using ANOVA and Chi² tests. 

• Examined the performance metrics between the ML, DL, and hybrid models to identify the most successful model.

• Created a web application that allows users to interact with the most optimal model. 

• Suggested using a mixed machine-learning approach to identify online harassment.

Reviewer #4:The paper has been revised thoroughly; however, there are still a few comments:

Author response:

Thank you for acknowledging the thorough revisions made to our manuscript. We appreciate your detailed review and the additional comments you have provided. We are committed to addressing these points to further improve the quality of our paper. Your constructive feedback is invaluable, and we sincerely appreciate the time and effort you have dedicated to reviewing our work.

Concern 1.Figure 2: Ensure the same font family and size are used consistently across all text elements for uniformity.

Author response:

Thank you for your valuable feedback. We deeply appreciate the time and effort you have taken to review our manuscript. Regarding your concern about the font consistency in Figure 2, we have made the necessary adjustments to ensure that the same font family and size are used consistently across all text elements for uniformity, with only the headlines being bold. Please let us know if there are any further adjustments or concerns. Thank you once again for your time and insights.

Author Action:

Concern 2. Figure 5: Comments before preprocessing should also be translated into English for the reader to understand the preprocessing.

Author response:

Thank you for your valuable feedback. We appreciate your thorough review of our manuscript. Regarding your concern about Figure 5 (note: revised menuscriptFigure 6), we have carefully translated all comments before preprocessing into English. This adjustment aims to enhance clarity and ensure that readers can fully understand the preprocessing steps involved. We hope these changes meet your expectations. Thank you once again for your valuable insights and guidance.

Author Action:

Fig 6. Original and processed text of Bengali Text Dataset.

Concern 3. The basic details of the methods, such as Stochastic Gradient Descent, Logistic Regression, Multi-layer Perceptron, Random Forest, Bagging Classifier, Adaptive Boosting, Deep Neural Network, Recurrent Neural Network, Convolutional Neural Network, Long Short-Term Memory, and Bidirectional Gated Recurrent Unit, seem unnecessary and make the paper lengthy. Consider condensing this section to focus on the most relevant methods to streamline the paper. Figures 10 and 11 also seem unnecessary.

Author response:

Thank you for your valuable feedback. We deeply appreciate the time and effort you have taken to review our manuscript. Regarding your concern about the detailed descriptions of the methods, we have made this section smaller and edited all method details. We believe that these concise descriptions are essential for ensuring clarity and comprehension of the methodologies used in our work. However, we have removed Figures 10 and 11 as per your suggestion, as they were deemed unnecessary.

If these changes do not meet your expectations, please let us know. We are committed to making any further adjustments required to improve the manuscript. Thank you once again for your insightful and constructive feedback.

Again, thank you for your kind cooperation and consideration.

Best Regards,

Khandaker Mohammad Mohi Uddin 

Corresponding Author

---

## [Editor Report · Decision Letter 2]

1 Aug 2024

Machine Learning and Deep Learning-Based Approach to Categorize Bengali Comments on Social Networks Using Fused Dataset

PONE-D-24-03168R2

Dear Dr. Mohi Uddin,

We’re pleased to inform you that your manuscript has been judged scientifically suitable for publication and will be formally accepted for publication once it meets all outstanding technical requirements.

Kind regards,

Muhammad Bilal

Academic Editor

PLOS ONE

---

## [Editor Report · Acceptance letter]

8 Aug 2024

PONE-D-24-03168R2 

PLOS ONE

Dear Dr. Mohi Uddin, 

I'm pleased to inform you that your manuscript has been deemed suitable for publication in PLOS ONE. Congratulations! Your manuscript is now being handed over to our production team.

Kind regards, 

on behalf of

Dr. Muhammad Bilal 

Academic Editor

PLOS ONE